METHODS AND RESOURCES

# Comprehensive characterization of the Hsp70 interactome reveals novel client proteins and interactions mediated by posttranslational modifications

Nitika[1]*, Bo Zheng[1], Linhao Ruan[2,3], Jake T. Kline[4], Siddhi Omkar[1], Jacek Sikora[5], Mara Texeira Torres[6], Yuhao Wang[2,3], Jade E. Takakuwa[1], Romain Huguet[7], Cinzia Klemm[6], Verónica A. Segarra[8], Matthew J. Winters[9], Peter M. Pryciak[9], Peter H. Thorpe[6], Kazuo Tatebayashi[10,11], Rong Li[2,3,12,13], Luca Fornelli[4], Andrew W. Truman[1]*

1 Department of Biological Sciences, The University of North Carolina at Charlotte, Charlotte, North Carolina, United States America, 2 Center for Cell Dynamics and Department of Cell Biology, Johns Hopkins University School of Medicine, Baltimore, Maryland, United States America, 3 Biochemistry, Cellular and Molecular Biology (BCMB) Graduate Program, Johns Hopkins University School of Medicine, Baltimore, Maryland, United States America, 4 Department of Biology, University of Oklahoma, Norman, Oklahoma, United States America, 5 Department of Molecular Biosciences, Department of Chemistry, and the Feinberg School of Medicine, Northwestern University, Evanston, Illinois, United States America, 6 School of Biological and Chemical Sciences, Queen Mary University, London, United Kingdom, 7 Thermo Scientific, San Jose, California, United States America, 8 Departments of Biological Sciences and Chemistry, Goucher College, Baltimore, Maryland, United States America, 9 Department of Biochemistry and Molecular Pharmacology, University of Massachusetts Medical School, Worcester, Massachusetts, United States America, 10 Laboratory of Molecular Genetics, Frontier Research Unit, Institute of Medical Science, The University of Tokyo, Tokyo, Japan, 11 Department of Biological Sciences, Graduate School of Science, The University of Tokyo, Tokyo, Japan, 12 Mechanobiology Institute and Department of Biological Sciences, National University of Singapore, Singapore, 13 Department of Chemical and Biomolecular Engineering, Whiting School of Engineering, Johns Hopkins University, Baltimore, Maryland, United States America

* nnitika@uncc.edu (N); atruman1@uncc.edu (AWT)

**Data Availability Statement:** The raw MS data has been deposited to the Proteome Xchange database

## Abstract

Hsp70 interactions are critical for cellular viability and the response to stress. Previous attempts to characterize Hsp70 interactions have been limited by their transient nature and the inability of current technologies to distinguish direct versus bridged interactions. We report the novel use of cross-linking mass spectrometry (XL-MS) to comprehensively characterize the *Saccharomyces cerevisiae* (budding yeast) Hsp70 protein interactome. Using this approach, we have gained fundamental new insights into Hsp70 function, including definitive evidence of Hsp70 self-association as well as multipoint interaction with its client proteins. In addition to identifying a novel set of direct Hsp70 interactors that can be used to probe chaperone function in cells, we have also identified a suite of posttranslational modification (PTM)-associated Hsp70 interactions. The majority of these PTMs have not been previously reported and appear to be critical in the regulation of client protein function. These data indicate that one of the mechanisms by which PTMs contribute to protein function is by facilitating interaction with chaperones. Taken together, we propose that XL-MS analysis of chaperone complexes may be used as a unique way to identify biologically important PTMs on client proteins.

(PXD036849). All other relevant data are within the paper and its Supporting Information files.

**Funding:** This work was supported by the NIH (R15GM139059 and R01GM139885 to AWT, R01GM057769 to PMP), the Queen Mary University of London and the Francis Crick Institute (Cancer Research UK—FC001183; UK Medical Research Council—FC001183 and the Wellcome Trust—FC001183 to PHT), a grant from Re-Stem Biotech to R.L., the JSPS Grants-in-Aid for Scientific Research (KAKENHI) (21H02422 to KT), the Institute for Fermentation, Osaka (G-2021-2-082 to KT). The funders had no role in study design, data collection and analysis, decision to publish, or preparation of the manuscript.

**Competing interests:** The authors have declared that no competing interests exist.

**Abbreviations:** AGC, automatic gain control; AP-MS, affinity purification mass spectrometry; BiFC, bimolecular fluorescence complementation; CID, collision-induced dissociation; CMV, human cytomegalovirus promoter; CTD, C-terminal domain; DIC, differential interference contrast; DSSO, disuccinimidyl sulfoxide; FA, formic acid; FPLC, fast protein liquid chromatography; GO, Gene Ontology; HIR, histone regulator; HSE, heat shock response element; HSP, heat shock protein; MS, mass spectrometry; NBD, nucleotide-binding domain; NCE, normalized collision energy; PTM, posttranslational modifications; SBD, substrate-binding domain; VC, Venus carboxy-terminal end; VN, Venus amino-terminal end; XL-MS, cross-linking mass spectrometry; Y2H, yeast two-hybrid.

- In vivo confirmation of Hsp70 dimerization

- Novel direct protein interactors of Hsp70

- Multidomain association between Hsp70 and its interactors

- Identification of novel biologically important client protein PTMs

## Introduction

The maintenance of a correctly folded proteome (proteostasis) is critical for cell survival. Cells maintain proteostasis under both basal and stress conditions through the expression of molecular chaperones such as Hsp70 and its associated co-chaperone regulators [1,2]. Hsp70 function is dependent on 3 conserved domains: an N-terminal nucleotide-binding domain (NBD), a substrate ("client")-binding domain (SBD), and a C-terminal "lid" domain (CTD) [1,2]. The binding and hydrolysis of ATP to ADP in the NBD promotes large-scale structural Hsp70 rearrangements that allow the closing of the CTD over client proteins that bind in the SBD, promoting protein folding [1,3]. The characterized roles of Hsp70 include folding of new and denatured proteins, transport of mitochondrial proteins and disaggregation of protein complexes [4–6].

The essential nature of Hsp70 function in the cell, as well as its involvement in a variety of human pathologies such as cancer, has driven researchers to set out to characterize Hsp70 interactors. While great strides have been made towards this goal, these efforts have been hampered by limitations in the technologies used. For example, these past efforts have utilized affinity purification followed by mass spectrometry (AP-MS), yeast two-hybrid (Y2H), and proximity proteomics methodologies, all of which lack the ability to discriminate between direct and bridged protein interactions [7–12]. Chemical cross-linking with mass spectrometry (XL-MS) is a powerful interactomic technique that circumvents this issue, providing information on direct interactions in protein complexes by using chemical cross-linkers [13–16]. Indeed, XL-MS studies are often complementary to traditional structural biology methods such as X-ray crystallography, nuclear magnetic resonance, and cryo-electron microscopy [16].

Importantly, Hsp70 stabilizes and activates of a wide range of signaling molecules including those involved in processes such as DNA damage response, cell cycle control, autophagy, and nutrient sensing [10,17–19]. The Hsp70 client proteins involved in these cellular processes tend to be either highly posttranslationally modified (PTMs) or regulate PTMs on other proteins. In turn, these PTMs tightly regulate a multitude of protein properties including subcellular localization, enzymatic activity, and protein interactions [20]. Advances in mass spectrometry-based methods have allowed the identification of more than 200 different types of PTMs on proteins including phosphorylation, acetylation, and ubiquitination [21–23]. Given the numerous PTMs identified on proteins, researchers are now facing difficult choices when selecting specific PTMs for further study. Computational methods for identifying important PTMs on proteins have been partially successful but rely on preexisting MS data [21,24]. In this report, we have utilized XL-MS to comprehensively understand the Hsp70 interactome. In doing so, we have uncovered not only a new set of interactors that directly bind to Hsp70, but also show that these proteins bind at multiple sites on Hsp70, including the N-terminal NBD. Notably, many of the Hsp70 interactions are in close proximity to novel biologically important PTMs. All in all, our data suggest that our XL-MS approach to chaperone

interactome characterization can also be used as a novel way to identify biologically important and previously unknown PTMs on proteins.

## Results

### Analysis of cross-linked yeast Hsp70 complexes

Previous studies have identified proteins in complex with yeast Hsp70 (Ssa1) using quantitative AP-MS [9]. To comprehensively identify novel direct Ssa1 interactors along with their associated points of interaction, we took a novel cross-linking proteomics approach. HIS-tagged Ssa1 was expressed in *ssa1-4Δ*, a yeast strain in which all 4 *SSA* (Hsp70) genes have been deleted [25]. HIS-Ssa1 complexes were cross-linked with disuccinimidyl sulfoxide (DSSO), an MS-cleavable cross-linker [26]. These complexes were digested into peptides via trypsin and were characterized via mass spectrometry (see Fig 1A). The DSSO cross-links present on the cross-linked peptides were cleaved by collision-induced dissociation during the MS2 stage of mass spectrometry, allowing analysis and identification of each of the single peptide chains at the MS3 level as in [26]. Overall, we identified 1,510 interactors associated with HIS-Ssa1 in the cross-linked complexes and 1,152 in control HIS-Ssa1 complexes without DSSO-mediated cross-linking (Fig 1B). We anticipated that proteins present in the DSSO-treated complexes may consist of direct Ssa1 interactors including client proteins and co-chaperones. To distinguish direct interactors of Ssa1 from indirect interactors, we filtered our data for cross-linked peptides where at least one of the identified peptides was Ssa1. We identified a total of 363 Ssa1-containing cross-linked peptides, out of which 177 were Ssa1-interactor cross-links and 106 were Ssa1-Ssa1 cross-links (Fig 1C). Validating our methodology, no cross-linked peptides were observed in the control uncross-linked sample. To determine whether the cross-linking process had enriched for any particular class of protein, we performed Gene Ontology (GO) analysis of unique candidate interactors of cross-linked and control samples. This GO analysis revealed enrichment of multiple cellular functions (Fig 1D). In the cross-linked samples, proteins involved in protein folding, trafficking, and cell signaling were all enriched, consistent with the established roles of Hsp70.

### Ssa1 oligomerization is required for a subset of chaperone functions

Nearly a third of the cross-linked peptides detected in our experiment were between 2 Ssa1 peptides (Fig 1C). To distinguish whether the cross-linked Ssa1-Ssa1 peptides came from dimerized Ssa1 molecules as opposed to single intramolecular cross-links, we initially mapped the identified cross-links onto homology-based Ssa1 models. Given the large conformational change Hsp70 undergoes during its folding cycle, we mapped obtained cross-links onto both ADP-bound ("open") and ATP-bound ("closed") structures of Hsp70 (Figs 2A and S1A). The mapping of cross-links onto these models revealed that a substantial number of Ssa1-Ssa1 peptides had cross-linking lengths well within the spacer arm limit for DSSO, while many others exceeded the lengths possible by cross-linking within a single molecule, implying potential dimerization (Figs 2B and S1A). Dimers of Hsp70 from different organisms have previously been observed [27] and we decided to evaluate whether the same was true for yeast Ssa1. To visualize Ssa1-Ssa1 interaction in live yeast, we utilized bimolecular fluorescence complementation (BiFC). Yeast expressing Ssa1 tagged with Venus amino-terminal end (VN) and Venus carboxy-terminal end (VC) were examined using high-resolution fluorescence microscopy. Imaging of these cells revealed that Ssa1 dimers were clearly visible and that they localized primarily to the nucleus, whereas no BiFC signal was observed in cells expressing only VN or VC-Ssa1 (Figs 2C and S1B). Previous structural studies of both the DnaK dimer in bacteria and Hsc70 dimer in mammalian cells identified 2 key residues important in dimer formation,

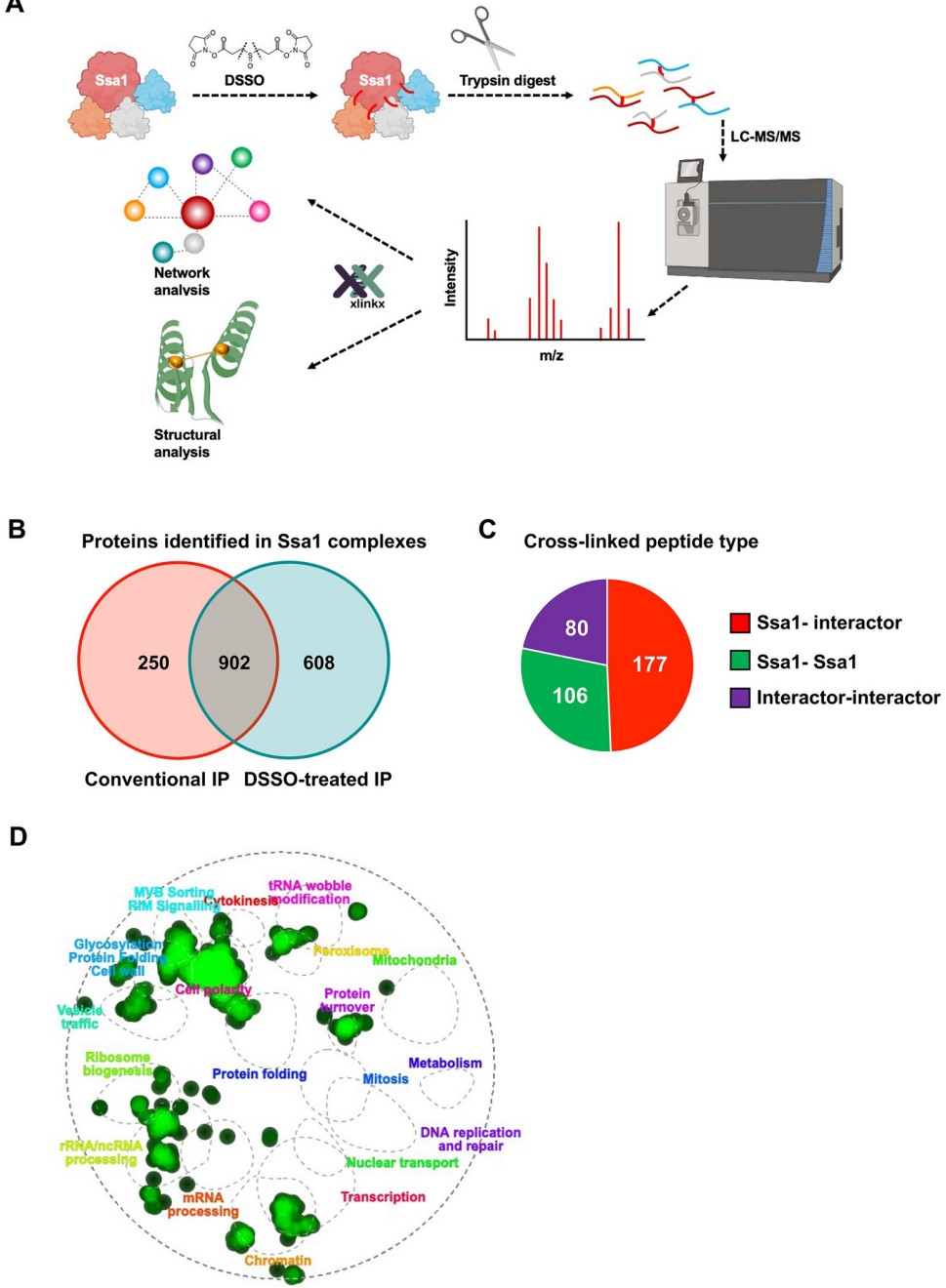

**Fig 1. Cross-linking mass spectrometry of Ssa1 complexes.** (A) Experimental workflow of cross-linking mass spectrometry of Ssa1 complexes purified from yeast cells. HIS-Ssa1 complexes were cross-linked with DSSO, purified from yeast, digested into peptides via trypsin, and analyzed by mass spectrometry. Created with BioRender.com. (B) Venn diagram representing Ssa1 complexes found in conventional IP and DSSO-treated IP. (C) Pie chart showing types of cross links identified from XL-MS analysis. (D) GO analysis of DSSO-treated Ssa1 immunoprecipitated complexes and cross-linked Ssa1 complexes using TheCellMap.org. DSSO, disuccinimidyl sulfoxide; GO, Gene Ontology; XL-MS, cross-linking mass spectrometry.

N537 and D540 [28,29]. Mutation of these sites on DnaK (N537A and D540A) results in bacteria that are heat sensitive [28]. We created equivalent mutations in Ssa1 (N537A/E540A) and assessed the ability of WT and the mutant to interact via co-immunoprecipitation (Fig 2D).

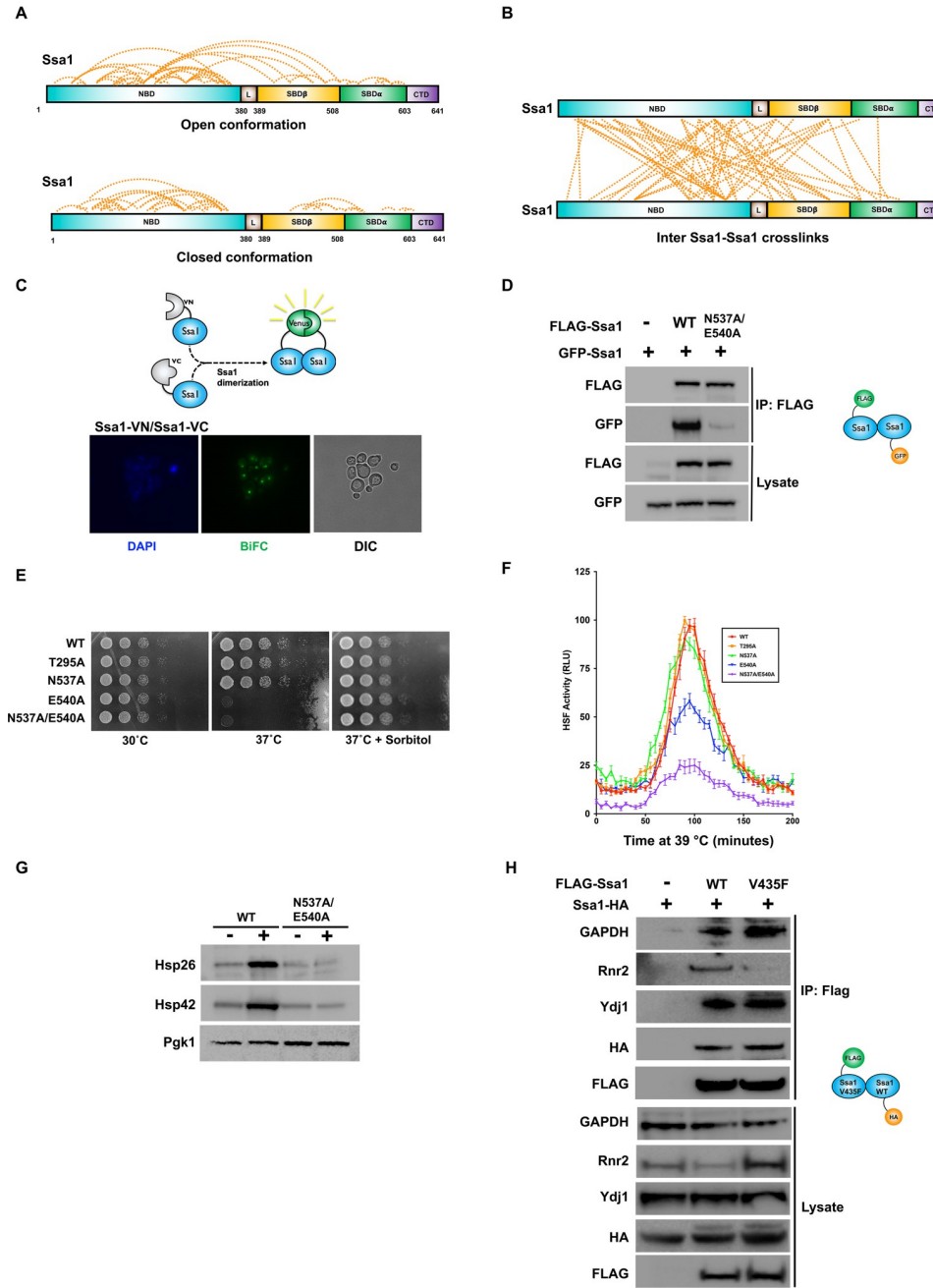

**Fig 2. Ssa1 oligomerization is required for a fully functional heat shock response.** (A) Identified Ssa1-Ssa1 cross-links mapped on the domains of Ssa1 in open and closed conformation. (B) Ssa1-Ssa1 cross-links that exceeded the DSSO spacer arm length when mapped to the monomeric structure of Ssa1 and thus potentially represent interactions between different Ssa1 molecules (dimers or oligomers). (C) Fluorescence images of diploid cells expressing both of the N-terminally VN- and VC-tagged versions of Ssa1. DAPI was used as a nuclear marker and the scale bars are 10 μm. (D) Analysis of Ssa1-Ssa1 interactions in yeast via co-immunoprecipitation. *Ssa1-4Δ* cells transformed with plasmids expressing GFP-Ssa1 and FLAG-Ssa1 were grown to mid-log phase. After extraction of total protein, FLAG-tagged Ssa1 complexes were purified via FLAG-magnetic beads and analyzed via SDS-PAGE/western blotting with indicated antisera. (E) Serial dilution of yeast expressing mutations that impact Ssa1-Ssa1 interactions. Yeast strains were grown to mid-log phase and then 10-fold serially diluted onto YPD media at the indicated conditions. Plates were photographed after 3 days. (F) Real-time luciferase reporter assay of Hsf1 activity over a 200-min heat shock at 39°C. Indicated yeast strains were transformed with a real-time luciferase reporter (HSE-lucCP+) and were processed as in [30]. The data shown are the average and standard deviation of at least 5 biological replicates. (G) Inducibility of Hsp26 and Hsp42 in WT and N537A/E540A yeast. Cells were grown to mid-log at 25°C and were then shifted to 39°C for

90 min. Protein lysate from these samples were analyzed by SDS-PAGE followed by western blotting using antisera for Hsp26, Hsp42, and Pgk1. (H) Western blot analysis of Flag-Ssa1 complexes (WT and V435F mutant) purified from cells expressing HA-tagged Ssa1. The data underlying the graphs shown in the figure can be found in S1 Data. CTD, C-terminal domain; DSSO, disuccinimidyl sulfoxide; NBD, nucleotide-binding domain; SBD, substrate-binding domain; VC, Venus carboxy-terminal end; VN, Venus amino-terminal end.

Similar to previous studies using bacterial DnaK or mammalian Hsc70, the ability of N537A/E540A mutant to interact with WT Ssa1 was substantially compromised (Fig 2D). To demonstrate in vivo functionality of the Ssa1 dimer, we examined the ability of N537A/E540A to support viability and the cellular response to heat. While yeast cells expressing the N537A/E540A dimer-deficient mutant were viable and grew at approximately WT rates, they were impaired for growth at high temperature (Fig 2E). The temperature-sensitive phenotype of N537A/E540A was suppressed by osmotic stabilization, suggesting impairment of the heat shock response and cell integrity signaling (Fig 2E). To further explain this temperature-sensitive phenotype, we assessed the activity of the heat shock response in WT and N537A/E540A cells using a real-time destabilized luciferase reporter [30]. WT cells produced a robust heat shock response element (HSE)-luciferase signal after heat exposure, whereas N537A/E540A cells did not (Fig 2F). To complement our HSE-reporter result, we determined the impact of N537A/E540A on the induction of 2 well-characterized heat-inducible proteins, Hsp26 and Hsp42. Hsp26 and Hsp42 levels did not respond to exposure to 39°C in N537A/E540A cells, confirming that loss of Ssa1 self-association impacts the heat shock response (Fig 2G). Insofar as Ssa1 is a major hub for protein folding in yeast, we set out to examine the possibility that some of the observed Ssa1-Ssa1 interactions might be the result of active Ssa1 folding a newly synthesized Ssa1 polypeptide chain. We studied the interactions of FLAG-Ssa1 (WT and substrate-binding deficient mutant V435F) with a known client, Rnr2 [9], GAPDH, a known interactor but not a client of Ssa1 in yeast [11], the Ydj1 co-chaperone, and HA-Ssa1. Although WT Ssa1 co-purified with GAPDH, Rnr2, Ydj1, and Ssa1, the V435F mutant maintained only interaction with GAPDH Ydj1 and Ssa1, demonstrating that Ssa1 is not a client of other Ssa1 molecules (Fig 2H). Taken together, these findings confirm that Ssa1 dimerizes in yeast and that this self-interaction is important for a subset of Ssa1 functions.

## Protein interactions of Hsp70 are distributed throughout its domains

Our Hsp70 cross-linking strategy identified 124 new direct interactors of Ssa1 (S1 Table and S2A Fig). Unique direct binding proteins identified using XL-MS were mapped on the domain structure of Hsp70 (Fig 3A). We detected interactions on 58% of the DSSO-accessible lysines (Fig 3A). Interestingly, in contrast to the established paradigm that Hsp70 client proteins bind and interact solely at the SBD, the majority (79%) of the identified direct interactions mapped to the NBD (Figs 3A and S2B). Given that DSSO cross-links lysines, we considered that an explanation for such a high number of interactions with the Ssa1 NBD may be explained by the number and distribution of lysines present in each domain. Even when accounting for the relatively large number of cross-linkable lysines, the NBD bound over 6 times the number interactions per cross-linkable lysine compared to the SBD (S2C Fig). While many of the NBD direct interactors relate to expected cell processes such as translation, chromatin organization, and protein folding, there are several with unknown biological functions (Fig 3B). To validate our XL-MS screen, we confirmed several of our hits using co-immunoprecipitation and immunoblotting. Consistent with our MS data, Cct8, Pcl7, Ura8, and Sse1 all co-purified with Ssa1 and associated chaperones/co-chaperones Sse1, Hsp82, and Ydj1 (Fig 3C).

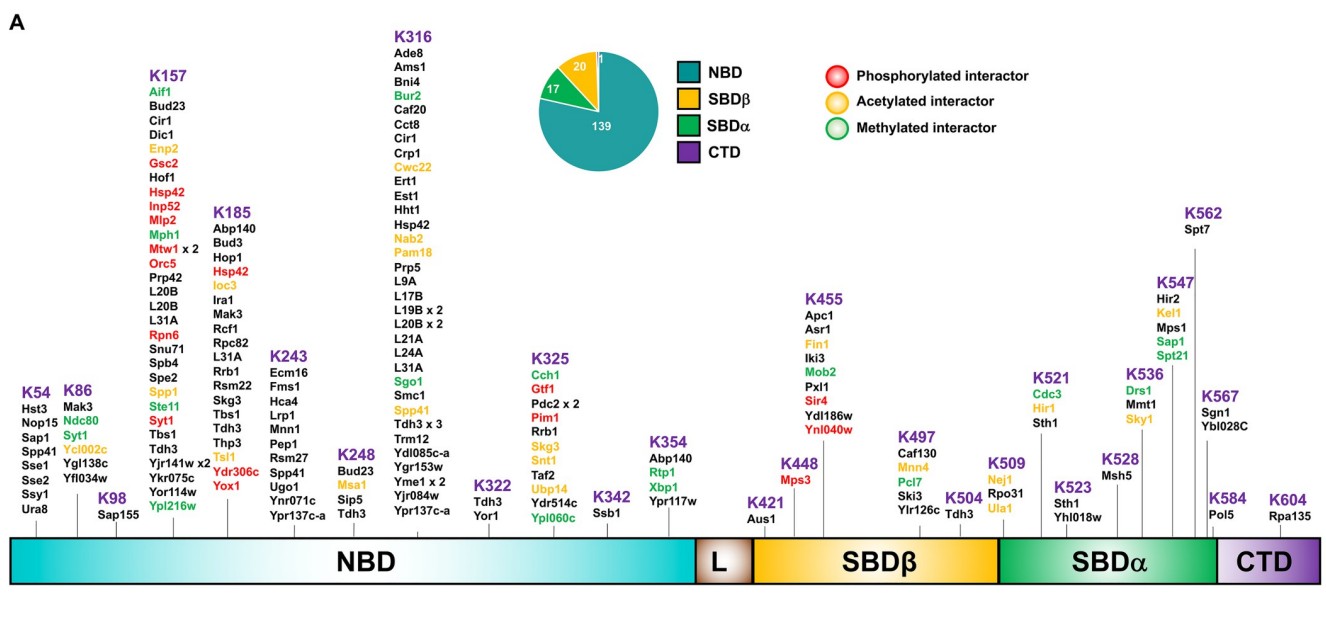

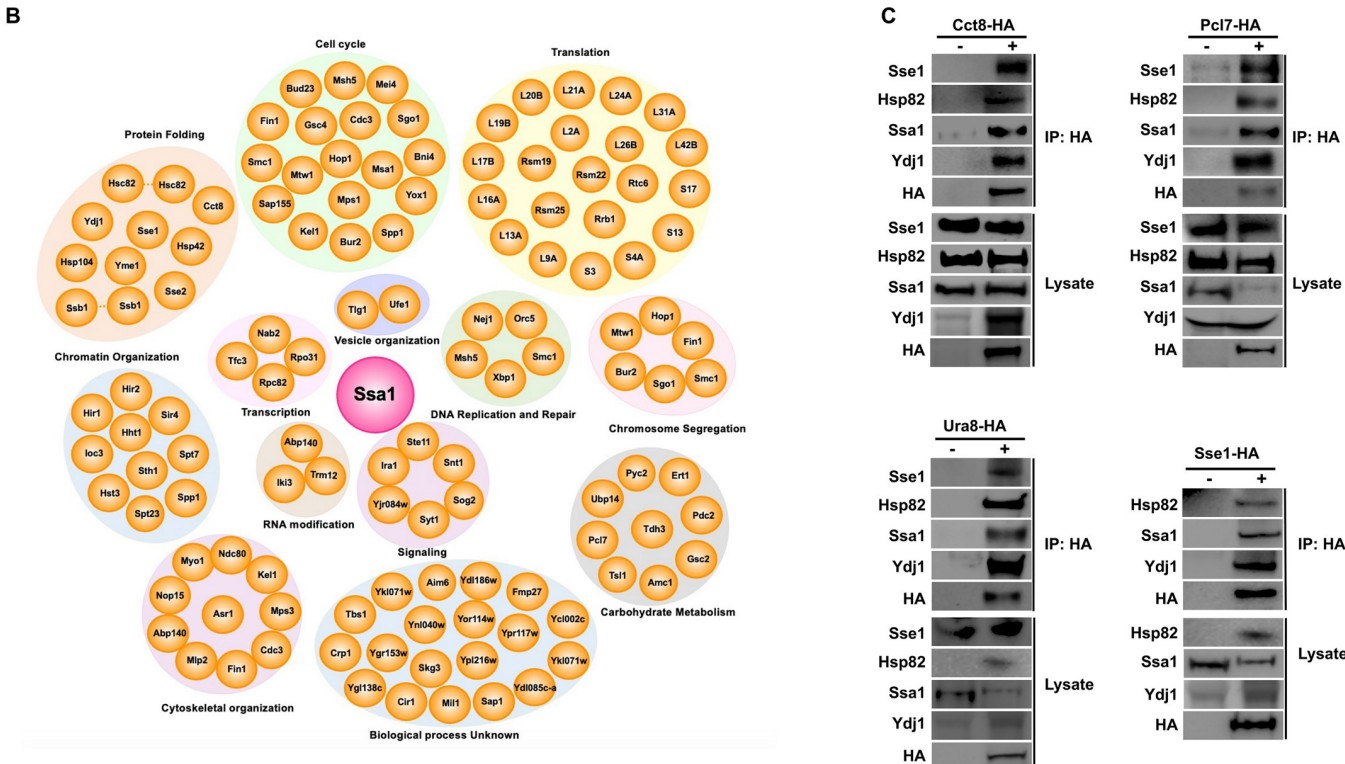

**Fig 3. Direct interactors of yeast Hsp70 identified by XL-MS.** (A) Schematic representation of 177 inter protein cross-links and identified PTMs of direct interactors on domains of Hsp70. (B) Functional classification of direct Hsp70-protein peptides. (C) Western blot analysis of HA-tag immunoprecipitated Cct8, Pcl7, Ura8, and Sse1 from yeast cells. CTD, C-terminal domain; NBD, nucleotide-binding domain; PTM, posttranslational modifications; SBD, substrate-binding domain; XL-MS, cross-linking mass spectrometry.

## The HIR complex is a novel client of Hsp70

The histone regulator (HIR) protein complex regulates histone gene transcription, nucleosome formation, and heterochromatic gene silencing [31,32]. Our XL-MS analysis revealed a novel

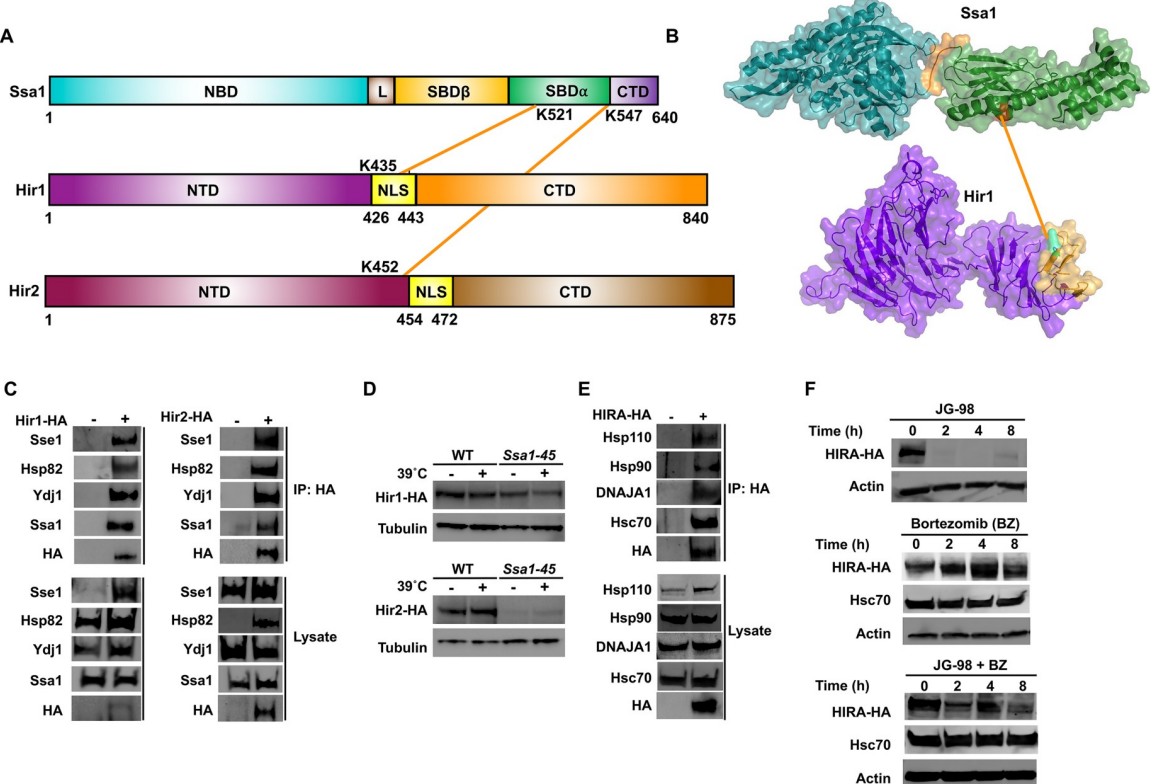

**Fig 4. HIR complex is a novel client of Hsp70 in yeast and humans.** (A) Schematic representation of Ssa1-Hir1/Hir2 inter protein cross-links detected on SBD of Ssa1 and NLS of Hir1 and NTD of Hir2. (B) Ssa1-Hir1/2 cross-links mapped on the crystal structure of Ssa1, Hir1, and Hir2. (C) The Hir complex interacts with yeast chaperones. (D) Hir1 and Hir2 are dependent on Ssa1 for their stability. Indicated yeast cells were transformed with plasmids expressing HA-Hir1 or HA-Hir2 driven via the *GAL1* promoter. Yeast were grown to mid-log in YPGalactose-URA media and then were either left untreated or were exposed to heat shock at 39˚C for 90 min. Levels of Hir1 or Hir2 were assessed via western blot using antisera to indicated proteins. (E) HIRA interacts with chaperone complexes in mammalian cells. HEK293 cells were transfected with an HA-HIRA construct. After 24 h, total protein was extracted and HIRA complexes were purified via HA-magnetic beads. The purified HIRA complexes were analyzed by SDS-PAGE followed by western blotting using the indicated antisera. (F) Western blot analysis of HIRA upon addition of Hsp70 inhibitor JG-98 and proteasomal inhibitor bortezomib. CTD, C-terminal domain; HIR, histone regulator; NBD, nucleotide-binding domain; SBD, substrate-binding domain.

direct interaction between Ssa1 and HIR complex components Hir1 and Hir2. We observed cross-linking between the SBD of Ssa1 and residues K435 of Hir1 and K452 of Hir2, adjacent to their respective nuclear localization signals (Fig 4A and 4B). To validate our XL-MS finding, we carried out Co-IP followed by immunoblotting of Hir1 and Hir2 with key chaperone components Ssa1, Sse1, Hsp82, and Ydj1. These experiments confirmed a strong association between HIR and the chaperones tested (Fig 4C). Furthermore, we confirmed that Hir1 and Hir2 are bona fide Ssa1 client proteins by demonstrating that loss of Ssa1 function resulted in Hir1 and Hir2 destabilization (Fig 4D).

To demonstrate evolutionary conservation of the identified chaperone-HIR interaction, we examined the interaction between human Hsc70 and HIRA, the major HIR complex protein in human cells. Consistent with our results in yeast, HA-HIRA co-immunoprecipitated with Hsc70, Hsp110, Hsp90, DNAJA1 (Fig 4E). To examine the dependence of HIRA on Hsc70 chaperone activity, we treated HEK293 cells with the Hsp70 inhibitor JG-98 and monitored HIRA abundance over time. HIRA levels rapidly decreased after JG-98 addition, with HIRA becoming undetectable after 2 h (Fig 4F). Given that in our system HA-HIRA was expressed

under the constitutive human cytomegalovirus (CMV) promoter, we hypothesized that the effect we observed on HIRA abundance could be explained by protein degradation. Supporting this hypothesis, addition of the proteasomal inhibitor bortezomib prevented JG-98-dependent HIRA loss (Fig 4F). Taken together, our results suggest that HIR complex proteins are client proteins of the Hsp70 chaperone system in yeast and mammalian cells.

## Exploring the biological importance of novel XL-MS-identified PTMs

Approximately 31% (55/177) of our cross-linked peptides contained a PTM such as acetylation, methylation, or phosphorylation (see S1 Table). A search for these PTMs using GPMDB (https://gpmdb.thegpm.org/) revealed that 95% of these PTMs had not been previously observed. After considering the possibility that these PTMs might be biologically important, we selected 3 different PTM-modified cross-links for further study from proteins in diverse cellular pathways; Pim1 (mitochondrial proteostasis), Mtw1 (chromosome segregation), and Ste11 (pheromone and osmotic stress response).

## Hsp70 plays a dual role in mitochondrial Pim1 protease activity

Pim1 is an ATP-dependent yeast Lon protease that is involved in degradation of misfolded mitochondrial proteins, required for mitochondrial maintenance and biogenesis [33]. Our XL-MS data revealed an interaction between the Pim1 protease domain and the N-terminal domain of Ssa1 (Fig 5A and 5B). We first validated the Pim1 interaction with Ssa1 and associated co-chaperones using co-immunoprecipitation and immunoblotting (Fig 5C). The clearance of mitochondrial aggregates is important for cell homeostasis and, as such, many organisms express a Pim1 homologue. To examine whether the Ssa1-Pim1 interaction is conserved in mammalian cells, we performed an equivalent experiment to that shown in Fig 5C, using mammalian Lonp-1 as the bait. Similar to our observations in yeast, mammalian Lonp-1 interacted with chaperone proteins including Hsc70, Hsp90, and DNAJA1 but not Hsp110 (Fig 5D). In order to determine if Lonp-1 is a client protein of Hsp70, we treated HEK293 cells with Hsp70 inhibitor JG-98 and observed Lonp-1 degradation after 2 h of treatment. Treatment of HEK293 cells with bortezomib before addition of JG-98 prevented loss of Lonp-1, confirming that Lonp-1 is a client protein of Hsc70 (Fig 5E).

The Pim1 portion of the Ssa1-Pim1 cross-linked peptide contained a previously undiscovered Pim1 phosphorylation site (S974). Given its proximity to the interaction surface between Ssa1 and Pim1, we hypothesized that this phosphorylation site might be important for Pim1 function. To examine the impact of S974 phosphorylation on Pim1 function, we expressed Pim1 mutants lacking this phosphorylation site (S974A) or mimicking constitutive phosphorylation (S974D) from the regulatable *CUP1* promoter in cells lacking Pim1. Although S974A cells grew at a similar rate to WT in standard growth media, S974D cells were substantially inhibited for growth (Fig 5F). To determine whether the growth defect of the S974D mutant was due to irregular mitochondrial protein aggregation, we examined the aggregation behavior of a previously established mitoFluc reporter [34,35]. S974D mutants were unable to clear mitochondrial protein aggregates (Figs 5G and S3A). To query whether the loss of Pim1 function in the S974D mutant was due to its mislocalization, we examined the localization of GFP-tagged WT, S974A, and S974D Pim1 proteins. Intriguingly, Pim1 localization was unaffected by the phosphorylation state of S974 (S3B Fig).

Like many proteases, Pim1 undergoes self-cleavage to achieve full maturation and protease activity [36–38]. We examined Pim1 processing in WT, S974A, and S974D cells. The protease inhibitor bortezomib binds to yeast, bacterial and mammalian forms of Pim1, suppressing its proteolytic activity [39–43]. In contrast to both WT and S974A, S974D resolved as a single

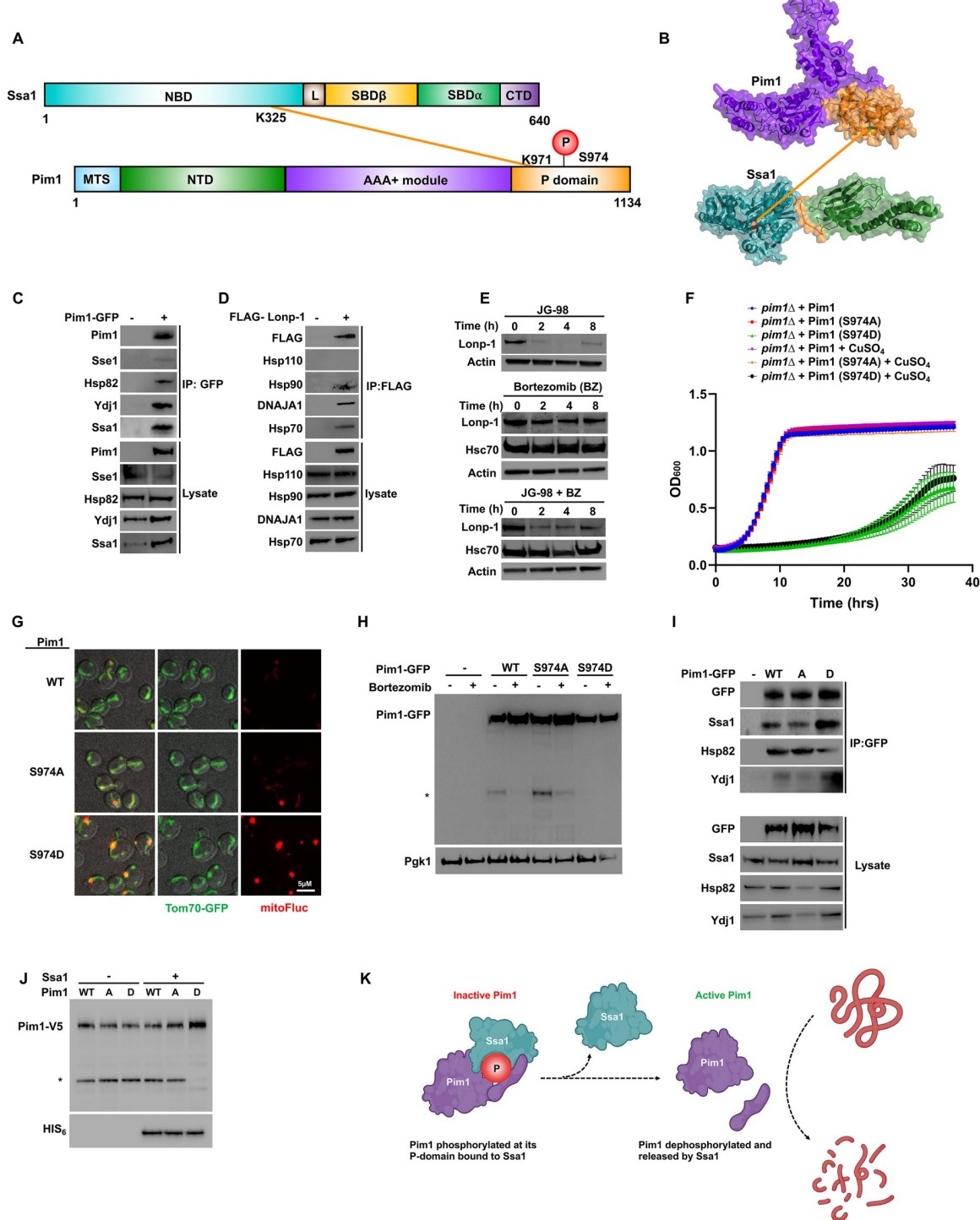

**Fig 5. Activity of Pim1/Lonp-1 is regulated by interaction with Hsp70 and a novel phosphorylation site, S974.** (A) Schematic representation of Ssa1-Pim1 inter protein cross-links detected on NBD of Ssa1 and proteolytic domain of Pim1. (B) Ssa1-Pim1 cross-links mapped on the crystal structures of Pim1 and Hsp70. (C) Pim1 interacts with the chaperone complex in yeast cells. Yeast expressing Pim1-GFP were grown to mid-log phase and Pim1-GFP complexes were isolated using GFP-TRAP beads. The purified Pim1-GFP complex was analyzed by SDS-PAGE/western blot analysis using indicated antisera. (D) Lonp-1 interacts with chaperone complexes in mammalian cells. HEK293 cells were transfected with a FLAG-Lonp-1 construct and after 24 h, total protein was extracted and FLAG-Lonp-1 complexes were isolated using FLAG Dynabeads. FLAG-Lonp-1 complexes were analyzed by SDS-PAGE/western

blot using indicated antisera. (E) Western blot analysis of Lonp-1 upon addition of Hsp70 inhibitor JG-98 and proteasomal inhibitor Bortezomib. HEK293 cells were grown to mid-confluence and were treated with the indicated reagents/times. Lysates were analyzed by SDS-PAGE/western blot using indicated antisera. (F) Growth assay of Pim1 phospho-site mutants in yeast. Indicated cells were grown under indicated conditions in 96-well format in a Synergy H1 plate reader. OD600 readings were taken at regular intervals for 35 min. Data shown are the average and standard deviation of at least 5 biological replicates. (G) Fluorescence images of cells expressing FlucSM–RFP and Tom70-GFP. Scale bars are 10 μm. (H) Western blot analysis of Pim1 wild type and phospho-mutants upon addition of Bortezomib. *Indicates Pim1 self-cleavage product. (I) IP analysis of Pim1 wild type and phospho-mutants with chaperone complex. Indicated cells were grown and processed as in (A). (J) Western blot analysis of Pim1 (wild type and phospho-mutants) expressed in the presence or absence of Ssa1 in *E. coli*. BL21 cells were co-transformed with indicated plasmids and were grown to early mid-log phase whereupon protein expression was induced with IPTG. After 4 h, total cell protein was isolated via sonication and lysates were analyzed by western blotting with indicated antisera. *Denotes Pim1 self-cleavage product. (K) Schematic of Ssa1 regulation of Pim1. Phosphorylated Pim1 interacts with Ssa1 preventing inappropriate activation of Pim1 in the cytoplasm. Pim1 dephosphorylation and Ssa1 dissociation permit Pim1 self-cleavage, critical for Pim1 activity in the mitochondria. Created with BioRender.com. The data underlying the graphs shown in the figure can be found in S1 Data. CTD, C-terminal domain; NBD, nucleotide-binding domain; SBD, substrate-binding domain.

band on SDS-PAGE, suggesting that Pim1 self-cleavage and maturation was compromised in S974D cells (Figs 5H and S3C). The proximity of S974 to the identified Ssa1-Pim1 cross-linked peptides suggested that this residue was important to the Ssa1-Pim1 interaction. Immunopre-cipitation of Pim1 variants demonstrated that, although not critical for Ssa1-Pim1 interaction, S974 phosphorylation significantly enhanced the interaction between the 2 proteins (Fig 5I).

Based on our results shown in Fig 5A–5I, we hypothesized that Ssa1 may bind to phosphorylated Pim1 to inhibit its function. To examine Pim1 activity in the absence of endogenous Ssa1 activity, we expressed recombinant yeast Pim1 (WT, S974A, and S974D) in *Escherichia coli* (Fig 5J). In contrast to our findings in Fig 5H, Pim1 self-cleavage was independent of S974 phosphorylation status in the absence of Ssa1 (Fig 5J). Upon co-expression of Pim1 and Ssa1 in bacteria, self-cleavage of Pim1 was prevented only in the S974D mutant (Fig 5J). Taken together, our findings suggest that fascinatingly, Pim1 is not only a novel client of Ssa1, but also that Ssa1 can prevent phosphorylation-mediated Pim1 self-cleavage (Fig 5K).

## Ssa1 regulates kinetochore function via Mtw1

Mtw1 is an essential component of the MIND kinetochore complex and it connects kineto-chore subunits binding DNA to those associated with microtubules, rendering it critical to kinetochore assembly [44]. Direct interaction between the NBD of Ssa1 and the head domain of Mtw1 was observed by XL-MS (Fig 6A and 6B). To confirm direct interaction between Mtw1 and the Hsp70 chaperone system proteins (Ssa1, Sse1, Hsp82, and Ydj1), we used a Co-IP and immunoblotting approach, similar to the one used above for other identified direct interactors (Fig 6C). Perturbation of Ssa1 function destabilized Mtw1 confirming its status as a new Hsp70 client protein in yeast (Fig 6D).

The mammalian equivalent of Mtw1, Mis12, is critical for correct kinetochore attachment [45]. The Ssa1-Mtw1 interaction is conserved in mammalian cells as evidenced by the success-ful co-purification of Mis12 with Hsc70, Hsp110, Hsp90, and DNAJA1 (Fig 6E). Furthermore, treatment of HEK293 cells with Hsp70 inhibitor JG-98 resulted in loss of Mis12, and addition of the proteasomal inhibitor bortezomib prevented JG-98-mediated Mis12 destruction, show-ing that inhibition of Hsp70 leads to proteasomal degradation of Mis12 (Fig 6F). Taken together, our results suggest that Mis12 is a novel client protein of Hsp70.

The identified site of interaction between Mtw1 and Ssa1 contained a previously undiscov-ered phosphorylation site, Y86 on Mtw1 (Fig 6A). Recent studies have suggested that Y86 is at the interface between Mtw1 and a second essential kinetochore subunit, Mif2 [46–48]. Haploid yeast cells expressing the non-phosphorylatable Y86F Mtw1 mutant protein were viable but exhibited an increased doubling time and were compromised for growth on the microtubule-

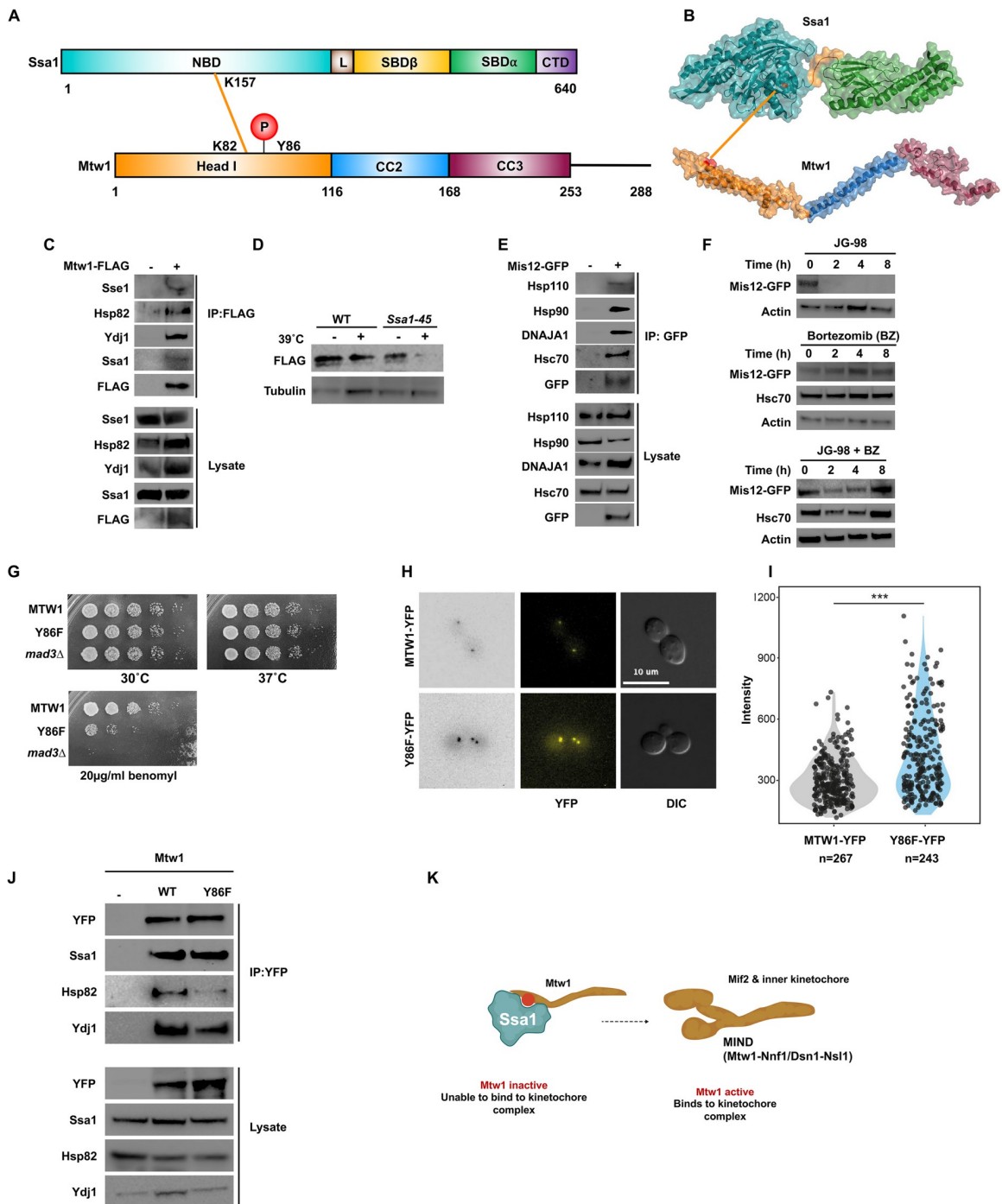

**Fig 6. Mtw1 is a client of Hsp70 and is regulated by phosphorylation.** (A) Schematic representation of Ssa1-Mtw1 inter protein cross-links detected on NBD of Ssa1 and head domain of Mtw1. (B) Ssa1-Mtw1 cross-links mapped on the crystal structure of Ssa1 and Mtw1. (C) Mtw1 interacts with the chaperone complex. Yeast expressing Mtw1-FLAG were grown to mid-log phase and Mtw1 complexes were isolated using FLAG dynabeads. The purified Mtw1 complex was analyzed by SDS-PAGE/western blot analysis using indicated antisera. (D) Mtw1 is destabilized in temperature-sensitive *Ssa1-45* mutant strain. Indicated yeast were transformed with a plasmid expressing Mtw1-GFP. Yeast were grown to mid-log and were then were either left untreated or were exposed to heat shock at 39°C for 90 min. Levels of Mtw1 were assessed via western blot using antisera to indicated proteins. (E) Mis12 interacts with chaperone complexes in mammalian cells. HEK293 cells were transfected with a Mis12-GFP construct and after 24 h, total protein was extracted, and Mis12 complexes were isolated using GFP-TRAP beads. Mis12 complexes were analyzed by SDS-PAGE/western blot using indicated antisera. (F) Western blot analysis of Mis12 upon addition of Hsp70 inhibitor JG-98 and proteasomal inhibitor bortezomib. (G) Growth assay analyzing the phenotype of the Mtw1 and its Y86 mutant. (H) Mtw1 was tagged with YFP in wild-type and Mtw1-Y86F mutant strains

to compare Mtw1 localization at the kinetochore using fluorescence microscopy. (I) Fluorescence intensities were quantified in wild-type and mutant Mtw1 using the semi-automated FociQuant ImageJ script [92]. Intensities were compared using the Student $t$ test ($p$-value = $1.8 \times 10^{-12}$). (J) Analysis of the impact of Mtw1 phosphorylation on interaction with chaperones. (K) Model of Mtw1 activity regulation via its phosphorylation. Dephosphorylation of Mtw1 promotes dephosphorylation and correct association with kinetochore components such as Mif2 and Nnf1. Created with BioRender.com. The data underlying the graphs shown in the figure can be found in S1 Data. CTD, C-terminal domain; NBD, nucleotide-binding domain; SBD, substrate-binding domain.

perturbing agent benomyl (Figs 6G, S4A and S4B). Notably, we were not able to generate cells expressing a phospho-mimetic Y86E Mtw1 mutant protein. We compared Mtw1 localization at the kinetochore in Mtw1-YFP (WT and Y86F) strains. Y86F cells showed a significantly increased accumulation of Mtw1 at the kinetochore compared to wild-type cells (Fig 6H and 6I). Although the Y86F mutation significantly impacted the interaction between Mtw1 and either Hsp82 or Ydj1, Ssa1 was unaffected by the Y86F mutation (Fig 6J). Taken together, our data suggests phosphorylation of Mtw1 and its interaction with Ssa1 affects its incorporation into kinetochores (Fig 6K).

## Ssa1 is involved in the selective activation of Ste11-mediated signaling

Ste11 is an MEK kinase involved in the cellular responses to both pheromone and hyper-osmolarity [49–52] (Fig 7A). It was an encouraging validation of our XL-MS methodology to observe direct interaction between the N-terminus of Ssa1 and the unstructured regulatory domain of Ste11, one of the first identified client proteins of Hsp90 (Fig 7B). As with Mtw1 and Pim1, the Ssa1-Ste11 peptide contained a previously undiscovered PTM, dimethylation on Ste11 R305. To determine the functional importance of dimethylation of Ste11 R305, we created the non-methylatable mutant R305A and dimethylation-mimic R305F and expressed these in cells lacking Ste11. To examine the impact of R305 on the pheromone response, we assessed the ability of Ste11 R305A and R305F to form halos in response to alpha factor, activate a FUS1-LacZ reporter, and promote Fus3 phosphorylation. In all 3 experiments, both R305A and R305F behaved in a similar manner to WT (Fig 7C–7E).

Although the R305 dimethylation status had minimal impact on the pheromone response, previous studies have demonstrated that some hyperactive mutations within this region (e.g., *STE11-Q301P* or *STE11-DDD*) rescue the osmo-adaptation defect caused by deletion of upstream Ste20 kinase upon high osmolarity [50,51]. We examined the ability of Ste11 R305 mutants to complement the loss of Ste11 in response to osmotic shock. Because Ste11 is essential for Hog1 MAPK activation by osmostress only when the other upstream pathway (the SLN1 branch) is inactivated, we deleted the *SSK2* and *SSK22* genes encoding the MEKKs for the SLN1 branch from the host cells Fig 7A. As with the pheromone response, the R305 dimethylation status appeared to be dispensable for Ste11 function in this regard (Fig 7F, upper panel). It is known that deletion of the upstream components of the Ste11-osmotic response pathway such as Ste20, renders cells sensitive to media containing NaCl [49–51]. Interestingly, we found that while expression of WT and R305A Ste11 in cells lacking native Ste11, Ssk2, Ssk22, and Ste20 had no discernible effect on osmotic resistance, R305F Ste11 rendered cells resistant to NaCl (Fig 7F, lower panel). To determine the impact of these R305 mutations on Hog1 activation, we performed reporter assays using the Hog1 reporter *8xCRE-lacZ* [50] on cells examined in Fig 7F. The R305F mutant was able to induce *CRE*-mediated transcription at a level several fold greater than WT Ste11, albeit at a lower level than that observed for previously characterized hyperactive Ste11 mutants DDD and Q301P (Fig 7G). To determine whether this R305F phenotype was specific to the osmotic stress response in *ste11Δ/ssk2Δ/ssk22Δ/ste20Δ* cells, we performed a halo assay on the cells from Fig 7G. In contrast to the results in Fig 7F and 7G, Ste11 R305F was indistinguishable from WT or R305A (Fig 7H).

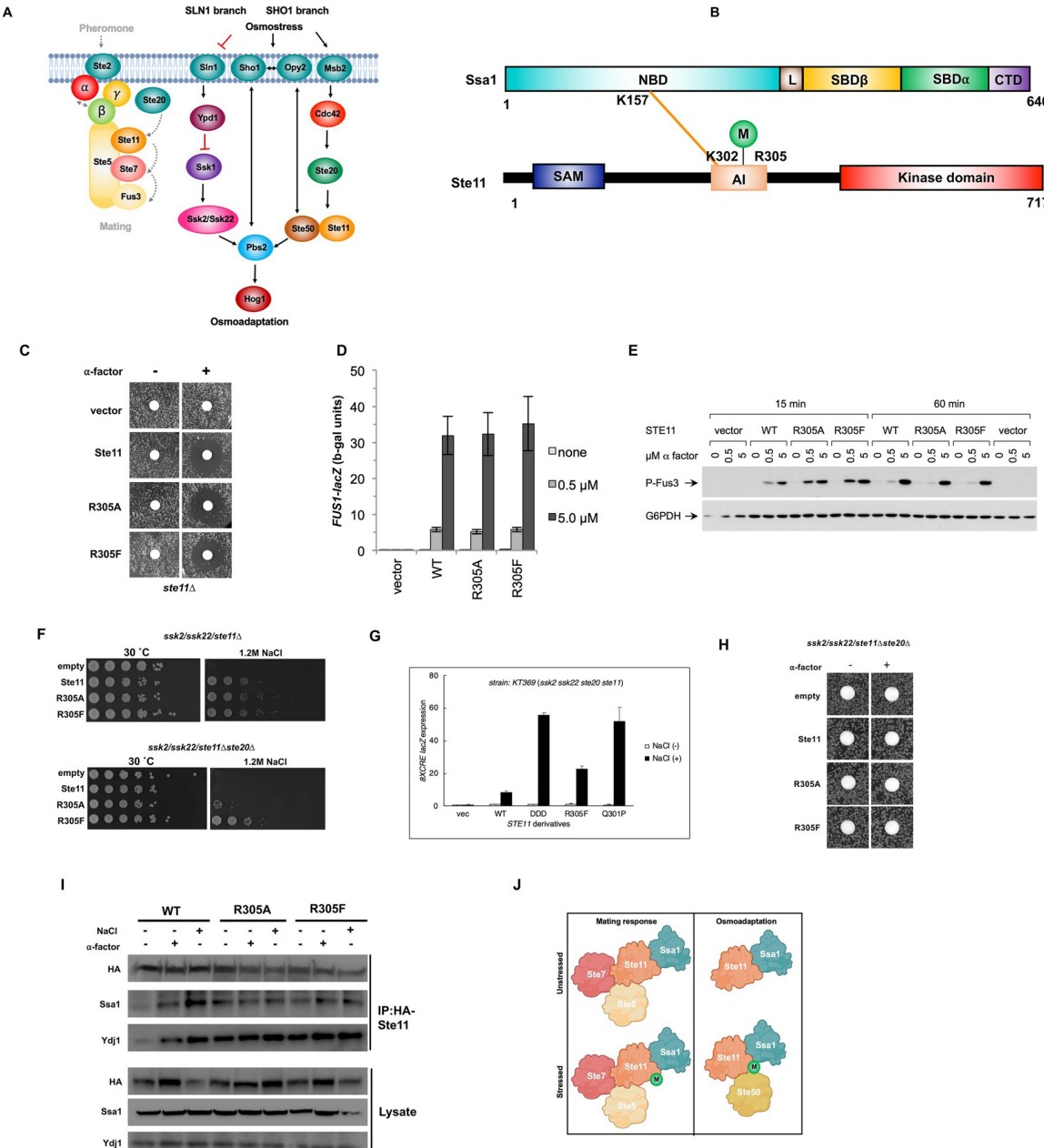

**Fig 7. Ste11 dimethylation impacts the osmotic stress response.** (A) Depiction of Ste11 pathway. (B) Schematic representation of Ssa1-Ste11 inter protein cross-links detected on NBD of Ssa1 and regulatory domain of Ste11. (C) Halo assay analyzing the phenotype of the Ste11 wild-type and methylation mutants in response to alpha factor. (D) FUS1-lacZ activity of Ste11 mutants in response to pheromone. Indicated yeast were grown to mid-log phase and then were processed as in [52]. (E) Western blot analysis of the effect of Ste11 wild type and mutants in response to pheromone signaling. Indicated cells were grown to mid-log phase, treated with the indicated quantity of alpha factor. Phosphorylation of Fus3 was measured by analysis of lysates via western blotting with indicated antisera. (F) Growth assay of Ste11 methylation mutants in response to hyperosmotic stress. Cells were grown to mid-log phase, were 10-fold serially diluted and then plated onto appropriate media using a 48-pin replica-plating tool. Images of plates were taken after 3 days at 30˚C. (G) 8xCRE-lacZ activity of Ste11 mutants in response to osmotic stress. Yeast transformed with the 8xCRE lacZ reporter were grown to mid-log phase and then treated with the indicated stressors. Cell lysate was extracted and beta-galactosidase activity was measured as in [50]. Data shown are the mean and standard deviation of at least 5 biological replicates. (H) Halo assay showing that none of the Ste11 variants permit pheromone response in the absence of Ste20. (I) Impact of Ste11 methylation on chaperone interactions. Yeast transformed with indicated HA-Ste11 constructs were grown to mid-log phase and Ste11 complexes were isolated using HA-magnetic beads. The purified Ste11 complexes were analyzed by SDS-PAGE/western blot analysis using indicated antisera. (J) Model of methylation-dependent regulation of Ste11 activity regulation. Di-methylation of Ste11 selectively alters osmotic signaling but not mating pathway activity. R405 di-methylation may alter Ste11 association with key osmotic signaling proteins such as Ste50. Created

with BioRender.com. The data underlying the graphs shown in the figure can be found in S1 Data. CTD, C-terminal domain; NBD, nucleotide-binding domain; SBD, substrate-binding domain.

We wondered whether R305 dimethylation might impact interaction with either Ssa1 or Ydj1, particularly given that the Ssa1-Ste11 cross-link was formed adjacent to the R305 site. Interaction studies using immunoprecipitated Ste11 suggest that R305 dimethylation was not essential for interaction with Ssa1 or Ydj1 (Fig 7I). Taken together, our data indicates a novel interaction of Ssa1 with the unstructured regulatory domain of Ste11 adjacent to a dimethylation site that impacts only osmotic signaling (Fig 7J).

## Discussion

### Towards a comprehensive Hsp70 interactome

The identification and characterization of new chaperone interactions is important to understand the fundamental process of protein folding [53]. This knowledge can ultimately lead to the design of novel therapies that rely on the manipulation of chaperone function. While large-scale interactome studies of chaperones have been attempted previously, the methods used in these attempts have associated drawbacks that prevent a comprehensive analysis of the system [54,55]. For example, several of these technologies such as LUMIER are performed on purified proteins and thus might not have the dynamic range to detect the impacts of PTMs and scaffold proteins [56,57]. Other cell-based assays such as AP-MS, Y2H and proximity labeling lack the ability to distinguish direct from bridged interactions. All of these drawbacks can be addressed by using the XL-MS technology we describe in this report.

XL-MS methodologies provide the ability to both stabilize transient interactions and allow characterization of the interaction surface between 2 proteins [13,16]. Recently, this innovative technology was used to obtain a more complete interactor list for Hsp90, providing a more accurate picture of its molecular dynamics [58]. For these reasons, we have used XL-MS to characterize the Hsp70 interactome in yeast. Our study identified a total of 1,510 different proteins complexed with Hsp70, 238 of which were cross-linked directly to Hsp70. Importantly, 121 of these direct interactions had never been previously observed, validating the use of XL-MS technology to comprehensively study the Hsp70 interactome in the future. It is interesting to speculate what the remaining 1,274 interactors represent. They may be bridged interactors present in association with Hsp70 complexes. If that is the case, it would suggest that large-scale datasets claiming to identify direct chaperone interactions might need to be revisited for accuracy. On the other hand, these interactions may be direct bona fide interactors that for technical reasons were unable to be directly cross-linked to Ssa1.

Previous in vitro studies suggested that the majority of interactions would be localized to the CTD of Hsp70, the domain recognized as being responsible for binding and processing of Hsp70 client proteins [59]. Using our XL-MS method, we identified new Ssa1 interactions that map to other Hsp70 domains (Fig 3A). Unexpectedly, 79% of the total interactions identified mapped to the NBD of Hsp70 (S1B and S1C Fig). Even after accounting for the number of cross-linkable lysines present on each Hsp70 domain, the NBD had over 6-fold the number of interactions compared to the CTD. Biologically, there may be several explanations for this result. Firstly, it is possible that during the client protein-binding process, there are multiple interactions between chaperone and client-protein that engage the entirety of Hsp70. Although this phenomenon has been observed in vitro between recombinant Hsp70 and single client proteins, our work is the first to make a similar observation at the interactome level [60]. Secondly, we may be detecting the interaction of Hsp70 in fully formed protein complexes, such

as observed in [61–63]. Finally, several of the N-terminal interactions may represent novel co-chaperones/regulators of Hsp70, which we hope to investigate in future studies. From a technical perspective, the imbalance of interactions observed between the N- and C-terminal domains of Ssa1 may arise from the cross-linker used. Hsp70 has a client-binding preference comprising of short stretches of amino acids enriched in hydrophobic and non-polar amino acids, with a model client peptide of NRLLLTG often being used in in vitro studies [64]. The cross-linker we used (DSSO) is a lysine cross-linker and it is possible that the lack of lysines on clients at the Ssa1 client-binding domain interface may prevent DSSO linkage [26]. It is also worth noting that the SBD on Ssa1 is heavily modified by PTMs, many of which occur on surface lysines [20]. The presence of these PTMs on the SBD may inhibit DSSO-cross linking at this region.

Our goal was to identify a more comprehensive interactome of Hsp70. To achieve this, we did not replenish ATP during the cross-linking and purification process, skewing the complexes towards co-chaperone free, client-bound Hsp70 complexes. In agreement with this, although several co-chaperones (Sse1, Cct8, Ydj1) were detected in complexes with Hsp70, very few were identified in our cross-linked samples. Although beyond the scope of this study, future experiments may entail purification of Hsp70 complexes in different stages of the folding cycle to trap co-chaperones rather than clients.

While proteomics methods can undoubtedly produce non-native interactions, all the hits selected for further study in this report were confirmed to be genuine Hsp70 interactors in both yeast and mammalian cells. Given the stress-dependent nature of the Hsp70 interactome, we hope to perform variations of this XL-MS experiment under different stress conditions such as heat, cell cycle stage, DNA damage response, and nutrient deprivation.

## Ssa1 self-association as a novel regulatory element of the heat shock response

An advantage of XL-MS methodologies is the ability to detect a wider range of information about protein folding and structure. For example, in the case of Hsp90, XL-MS has been used to understand protomer conformational changes upon ATP binding [58]. In this study, we identified 177 internal Ssa1-Ssa1 cross-links. Although the structure of full-length Ssa1 protein has yet to be obtained, sequence similarity to bacterial and mammalian Hsp70 strongly suggests that Ssa1 forms similar ATP and ADP-bound conformations [1,65,66]. Serving as an internal control to our experiment, the majority of obtained Ssa1-Ssa1 peptides could be matched to these structures. We were however surprised by the number of remaining peptides that could not be matched to any known monomeric Hsp70 structure. Deeper analysis of these peptides revealed that these were likely the result of cross-linking of *two different* Ssa1 molecules. The evidence supporting this is 2-fold. Firstly, the distance between these cross-linked Ssa1-Ssa1 peptides exceeded the known DSSO cross-linker length. Secondly, several of the cross-linked Ssa1-Ssa1 peptides were symmetrical; the peptides on each side of the cross-link were the same. While clearly not at the same stoichiometry as Hsp90, several studies on bacterial and human Hsp70 have demonstrated a capacity for the purified chaperone to form higher-order structures [28,29,61,65,67,68]. Expression of a dimerization-deficient DnaK in bacteria produces viable cells that are sensitive to thermal stress, suggesting that dimerization is needed for a subset of DnaK functions [67]. Through both co-immunoprecipitation and BiFC, we demonstrate for the first time that yeast Ssa1 can also self-associate in cells. These Ssa1 multimers are almost exclusively localized to the nucleus, suggesting a nuclear-specific function. Mutations that impair Ssa1 self-association result in yeast that while viable are unable to grow at high temperatures, activate an HSE-driven reporter or induce Hsp26 and Hsp42

upon heat shock. These data suggest that Hsp70 self-association is required for a full activation of the heat shock response. In accordance with this idea, the temperature sensitivity of N537A/E540A could be suppressed by sorbitol, phenocopying cells that have defects in Hsf1 activity [69]. Hsp70 is a well-characterized regulator of the heat shock response in yeast and mammalian cells [30,70–74]. It is interesting to speculate that the shift from monomer to dimer/oligomeric form may disrupt the interaction of Ssa1 with the heat shock transcriptional machinery. In future studies, we hope to further explore the stoichiometry, role, and formation of Ssa1 multimers, particularly with regards to the heat shock response.

## Using sites of Hsp70 interaction to reveal novel PTMs of functional relevance on client proteins

Interactions with molecular chaperones are critical for supporting the function of proteins involved in signal transduction, particularly those involved in PTMs such as kinases or acetylases [75–77]. Some of these interactions are quite stable, with client proteins requiring continuous chaperone interaction for activity [2]. Others are transient, where full client protein maturation and activity requires rapid chaperone dissociation [1]. However, there is also a growing body of work that shows that PTMs play an important role in mediating chaperone interactions. Chaperones are highly modified by a range of PTMs that includes phosphorylation, methylation, acetylation, and ubiquitination. These modifications, collectively known as the "Chaperone Code," can regulate many facets of chaperone function including their global interactions with both clients and co-chaperones [6,20,78]. In our study, all of the PTMs detected on our cross-linked peptides were on Hsp70 interactor side. Here, we have followed up on 3 Hsp70 clients that contain novel PTMs on the site of interaction with Ssa1: Mtw1, Pim1, and Ste11. In each of these cases, while the client protein PTMs had been previously undiscovered, we have shown them to regulate novel and distinct (and very different) facets of protein function.

In regards to Pim1/Lonp-1, this study establishes both as new clients of the Hsp70 chaperone system. While previous studies have demonstrated the capacity for Pim1 and Lonp-1 to cleave themselves to become active, there is minimal understanding of how this process is regulated [37,38]. Excitingly, here we have uncovered that Pim1 S974 (a site observed on the interface with Ssa1) regulates Pim1 self-processing and protease activity. In our in vitro experiments, we demonstrated that the phosphorylation-dependent self-processing of Pim1 is dependent on the presence of Ssa1. In building a model of the Ssa1-Pim1 interaction, it is important to consider their relative cellular localizations. Pim1 functions primarily in the mitochondria, whereas Ssa1 is present mostly in the cytoplasm. Our current working model is that Ssa1 interacts with phosphorylated Pim1 in the cytoplasm, preventing it from inappropriately cleaving cytoplasmic proteins. Release of active Pim1 from Ssa1 into the mitochondria would allow Pim1 to perform its intended function. We hope to investigate the kinetics and regulation of this phosphorylation site in future studies. Inhibition of Hsp70 function in mammalian cells promotes rapid degradation of Lonp-1. Alterations in Lonp-1 are implicated in an array of human pathologies that include cancer, neurodegeneration, heart disease, and stroke [79]. Manipulating interactions between chaperones and LonP-1 may form the basis of novel therapies for these illnesses.

For Mtw1, our XL-MS experiment revealed a novel phosphorylation site, Y86 that mediates its chaperone/kinetochore interactions. Loss of Mtw1 Y86 phosphorylation promoted altered increased Mtw1 kinetochore localization and Y86F cells were sensitive to benomyl-containing media. It may that this version of Mtw1 cannot be bound and folded properly by chaperones or cannot be cycled on and off the kinetochore when required. Importantly, structural studies

of the kinetochore place Y86 in close proximity to Nnf1 and Mif2 kinetochore proteins [46–48]. We were unable to create a phosphomimic version of Y68, possibly because this mutant becomes locked onto Ssa1/other chaperones and is unable to interact with Mif2/Nnf1. Taken together, our results suggest that phosphorylation of Mtw1 at Y86 may be dynamic, being added and removed as part of a chaperone-mediated assembly of the kinetochore.

Ste11 was one the first bona fide clients of Hsp90 identified and was thus exciting for us to have detected it as a direct interactor of Ssa1. The role of Hsp90 and co-chaperones in supporting Ste11 function is through stabilization of its kinase domain [80–83]. Several studies have connected Ssa1 to Ste11 pathway function, but its specific role remains unclear [84,85]. Our XL-MS data suggests that Ssa1 is binding adjacent to Hsp90 on the flexible linker region of Ste11 rather than the kinase domain. The di-methylation we identified on R305 has not been previously observed but exists close to known phosphorylation sites that are important for Ste11 activity [51,86]. Our working model is that dimethylation regulates the transition between inactive and active Ste11 conformations, either independent of or in conjunction with its regulation by phosphorylation. While beyond the scope of this study, we hypothesize that this dimethylation is selectively impacting interactions of Ste11 with components of the osmotic stress response pathway such as Ste50 or Ahk1, which bind close to this region of Ste11 [49].

Although work is well underway to tackle the complex nature of the chaperone code [20,87,88], the requirements for PTMs on client proteins to bind and be processed by chaperones is poorly understood. A small number of examples have been identified, including the Hsp90-Mpk1 interaction in yeast that is driven by the dual phosphorylation of Mpk1 [7]. While such PTM-associated interactions have been identified on a one-at-a-time basis, ours is the first study to identify PTM-associated chaperone interactions on a much larger scale. Over 95% of the PTMs identified on Ssa1 interactors in this study were previously unknown. This study suggests that perhaps our XL-MS methodology can be used to identify low-stoichiometry PTMs of high functional relevance on a wide range of proteins. We find it fascinating that such a large number of Ssa1 interactors contain PTMs at their site of interaction and suggests to us the potential for a "client code" that dictates protein interactions with chaperones in cells. Future studies will aim to decipher the stresses and enzymes that regulate these PTMs in addition to teasing apart their hierarchy of interaction with Ssa1.

This study has uncovered novel direct interactors of Hsp70, expanding its role in the cell to include chromatin regulation, kinetochore formation, mitochondrial protein processing, and MAPK signaling specificity. In doing so, we have established novel tools, methods, workflows, and datasets of novel PTMs that will allow the field to achieve a more complete understanding of chaperones and the ways in which cells use them to integrate signal transduction pathways.

## Methods

### Reagents and resources

Details on all reagents and resources (yeast strains and plasmids) are provided in S2 Table.

### Cell lines and cell culture

HEK293T cells were cultured in Dulbecco's modified Eagle's minimal essential medium (DMEM; Thermo Fisher Scientific) supplemented with 10% fetal bovine serum (Thermo Fisher Scientific), GlutaMAX (Thermo Fisher Scientific), 100 U/ml penicillin (Thermo Fisher Scientific), and 100 μg/ml streptomycin (Thermo Fisher Scientific). All cell lines were incubated at 37°C in a 5% $CO_2$-containing atmosphere.

### Yeast strains and growth conditions

Yeast cultures were grown in either YPD (1% yeast extract, 2% glucose, 2% peptone) or grown in SD (0.67% yeast nitrogen base without amino acids and carbohydrates, 2% glucose) supplemented with the appropriate nutrients to select for plasmids and tagged genes. *E. coli* DH5α was used to propagate all plasmids. *E. coli* cells were cultured in Luria broth medium (1% Bacto tryptone, 0.5% Bacto yeast extract, 1% NaCl) and transformed to ampicillin or kanamycin resistance by standard methods.

For tagging genomic copies of *CCT8*, *PCL7*, *URA8*, and *SSE1* with an HA epitope tag at the carboxy terminus, the pFA6a-HA-His3MX6 plasmid was used. For tagging the genomic copies of SSA1 for BiFC, pFA6a-VN-His3MX6 and pFA6a-VC-kanMX6 plasmids were used.

### Plasmid construction

The *SSA1* gene was PCR amplified from yeast genomic DNA using primers that resulted in overhangs containing SpeI and ClaI recognition sites on 5′ and 3′, respectively. This SpeI/ClaI-digested product was ligated into SpeI/ClaI-digested pUG36 (gift of U. Gueldener and J. H. Hegemann). The resulting plasmid expresses GFP-tagged Ssa1, the expression of which is driven via the MET25 promoter.

### Bacterial strains and growth conditions

*E. coli* DH5α was used to propagate all plasmids. The BL-21 strain was used for protein expression. *E. coli* cells were cultured in Luria broth (1% tryptone, 0.5% yeast extract, 1% NaCl) and transformed to carbenicillin resistance by standard methods. Plasmids are listed in Supporting information (S2 Table).

## Mass spectrometry

### Sample preparation and cross-linking

Yeast cells expressing His-tagged Ssa1 were grown to mid-log phase in SD-leu media. Cells were harvested and protein was extracted via bead beating followed by sonication. His-tagged Ssa1 was purified using an ÄKTA prime Plus fast protein liquid chromatography (FPLC) system (GE Healthcare) equipped with a 1-ml His-Trap HP column, followed by buffer exchange into dialysis buffer. Purified protein concentration was quantified by Coomassie assay (Thermo Fisher Scientific). Approximately 150 μg of purified His-Ssa1 complex was cross-linked with 5 mM DSSO and incubated at room temperature for 1 h. The reaction was quenched using 20 mM Tris-HCl (pH 8.0).

### In-solution sample digestion and SCX fractionation

Approximately 150 μg of cross-linked or uncross-linked samples were mixed with 6× volumetric excess of ice-cold acetone and precipitated overnight at −80°C. Precipitated proteins were pelleted at 21k RCF at 4°C and re-solubilized in 150 μL of 8M Urea/0.1 M$NH_4HCO_3$ reduced with 10 mM DTT for 30 min and alkylated with 50 mM IAA for 30 min in the dark. Samples were diluted 4× with 100 mM $NH_4HCO_3$ to reach 2M Urea concentration and digested with trypsin (trypsin/protein ratio of 1/50) overnight at 37°C. Resulting mixture of tryptic peptides was concentrated using SpeedVac and re-suspended in 10 mM $KH_2PO_4$ (pH 2.8), 20% ACN and loaded on preconditioned polysulfoethyl A (12 μm, 300 Å) solid phase cartridges. Peptides were eluted with increasing concentration (0, 4, 8, 12, 25, 50, 125, 250, and 500 mM) of KCl. Resulting fractions were desalted with Peptide Desalting Spin Columns (Thermo Fisher Scientific Pierce—89851) according to manufacturer's protocol, dried down on SpeedVac, and resuspended in 0.1% formic acid (FA).

## Liquid chromatography–tandem mass spectrometry peptide analysis

Resuspended cross-linked peptides were separated by nanoflow reversed-phase liquid chromatography (LC). An Easy-nLC 1000 (Thermo Fisher Scientific, San Jose, California, United States of America) was used to load approximately 1 µg of peptides on the column and separate them at a flow rate of 300 nl/min. The column was a 50-cm long EASY-Spray C18 (packed with 2 µm PepMap C18 particles, 75 µm i.d., Thermo Fisher Scientific, Sunnyvale, California, USA). The analytical gradient was performed by increasing the relative concentration of mobile phase B in the following steps: from 2% to 28% in 65 min, from 28% to 36% in 10 min, and from 32% to 90% in 5 min (for washing the column). The wash at high organic concentration was followed by re-equilibration of the column at 2% B for 10 min, for a total run time of 90 min. A shorter version of the gradient (total run time of 60 min, obtained by shortening the first step) was used for blanks and for standard shotgun proteomics targeting all peptides (not only cross-linked ones) present in the sample. Mobile phase A was composed of an aqueous solution of 0.1% FA, while mobile phase B consisted of 19.9% water, 80% acetonitrile, and 0.1% FA. A 2-kV potential was applied to the column outlet using an EASY-Spray nanoESI source (Thermo Fisher Scientific, San Jose, California, USA) for generating nano-electrospray.

All mass spectrometry (MS) measurements were performed on a tribrid Orbitrap Fusion Lumos (Thermo Fisher Scientific, San Jose, California, USA). For the identification of cross-linked peptides, a specific data-dependent acquisition method described by Liu and colleagues was applied [89]. Briefly, broadband mass spectra (MS1) were recorded in the Orbitrap over a 375 to 1,500 $m/z$ window, using a resolving power of 60,000 (at 200 $m/z$) and an automatic gain control (AGC) target of 4e5 charges (maximum injection time: 50 ms). Precursor ions were quadrupole selected (isolation window: 1.6 $m/z$) based on a data-dependent logic, using a maximum duty cycle time of 5 s. Monoisotopic precursor selection and dynamic exclusion (30 s) were applied. Peptides were filtered by intensity and charge state, allowing the fragmentation only of precursors from 4+ to 8+. Tandem mass spectrometry (MS2) was performed by fragmenting each precursor passing the selection criteria using both collision-induced dissociation (CID) with normalized collision energy (NCE) set at 25% and electron transfer dissociation—higher energy collisional dissociation (EThcD), with ETD reagent target set at 5e5, reaction time calculated on the basis of a calibration curve and supplemental collisional activation set at NCE = 15%. The AGC target for both CID and EThcD MS2 was set at 5e4 (maximum injection time: 100 ms), and spectra were recorded at 30,000 resolving power. CID MS2 spectra where a diagnostic neutral loss characteristic of the DSSO cross-linker (Δm = 31.9721 Da) was found between 2 pairs of product ions were used to trigger a data-dependent MS3 scan based on higher energy collisional dissociation (HCD) with NCE = 30%, using a multi-notched isolation (notch width = 2 $m/z$, 2 precursors selected), an AGC target of 2e4 and spectral detection in the linear ion trap (operating in rapid mode).

For the identification of all proteins included in the samples, the data-dependent acquisition method was simplified using uniquely HCD for tandem MS, recording MS2 spectra over a 110 to 2,000 $m/z$ window using 15,000 resolving power (at 200 $m/z$), and an AGC target of 2e4 (maximum injection time: 30 ms). Peptides with charge states from 2+ to 7+ were considered for fragmentation. MS1 scans were recorded at 120,000 resolving power (at 200 $m/z$).

## MS Data analysis

All data analysis was carried out with Protein Discover 2.2. For identification of cross-linked peptides CID/EThcD RAW data were searched with a crosslink processing workflow. For XlinkX, Detect node parameters were as follows: Acquisition strategy: MS2_MS3 with Crosslink Modification: DSSO/+158.004Da(K). Both XlinkX search node and in the SequestHT

nodes search an SGD orf FASTA database (SGD_orf_trans_2015-01-13—because of high degree of identity, we decided to keep only Ssa1 isoform of Ssa in searched database) and trypsin enzymatic specificity with 2 maximum missed cleavages. Precursor Mass Tolerance was 10 ppm and Fragment Mass Tolerance was 0.6 Da. Carbamidomethylation (C) was allowed as a static modification. Dynamic modifications were as follows: Oxidation(M), DSSO Hydrolyzed(K), DSSO Tris (K) Acetyl (protein N-term). Additionally, we searched data for phospho (S, T, Y), acetyl (K), and single, di, tri methylation (K) separately.

Cross-links were validated using the Percolator strategy and the FDR threshold was set to 0.01. Finally, results were filtered for high-confidence peptides using consensus step. Control peptide error rate strategy was used and 0.01 (strict) and 0.05 (relaxed) values were used for Target FDR for both PSM and Peptide levels. Only high-confidence peptides were included, and minimal peptide length was set to 5.

For general identification of all proteins included in the samples, HCD fragmentation data were processed with Protein Discoverer 2.2 utilizing Sequest HT and MS Amanda search engines. For both Precursor Mass Tolerance was 10 ppm and Fragment Mass Tolerance was 0.2 Da. Carbamidomethylation (C) was allowed as a static modification and dynamic modifications were as follows: Oxidation(M), Acetyl (protein N-term). Identified peptides were validated using Percolator and target FDR value was set to 0.01 (strict) and 0.05 (relaxed). Finally, results were filtered for high-confidence peptides using consensus steps. Control peptide error rate strategy was used and 0.01 (strict) and 0.05 (relaxed) values were used for Target FDR for both PSM and Peptide levels. A full table of Ssa1-Ssa1 cross links, Ssa1- client cross-links, and PTMs identified can be found in S1 Table. The mass spectrometry proteomic data have been deposited to the ProteomeExchange Consortium (http://proteomecentral.proteomexchange. org) via the PRIDE partner repository with the dataset identifier PXD001284.

## Bioinformatics analyses

### Gene ontology enrichment analysis

GO analysis of DSSO-treated Ssa1 immunoprecipitated complexes and cross-linked Ssa1 complexes were accomplished using TheCellMap.org.

### Protein interaction network analysis

Proteins identified in cross-linked dataset were analyzed as described previously [10]. All nodes correspond to identified proteins. Cytoscape analyses were represented as network maps. Crosslink distances for all the inter and intra protein cross-links were measured using PyMOL software (https://pymol.org/2/).

### General data analyses

Data processing and analyses were performed using GraphPad Prism (version 7).

### Growth assays

For serial dilutions, cells were grown to mid-log phase, 10-fold serially diluted and then plated onto appropriate media using a 48-pin replica-plating tool. Images of plates were taken after 3 days at 30˚C. For growth curves, yeast cells were grown to mid-log phase and optical density was measured at 600 nm for indicated times. For the induction of Pim1 under CUP1 promoter 500 μm $CuSO_4$ was added to YPD media for 3 h. Cells were then used for microscopy and immunoblotting.

## Yeast halo assay

For halo assay, yeast cells were grown to mid-log phase. The following day, the culture was diluted 1:1,000 and 150 μl of cells were spread onto a YPD plate. After the plate had been incubated for 2 h at 30˚C, 10 μl of 5 μg/ml of synthetic α factor peptide (WHWLQLKPGQPNleY) in DMSO was spotted onto circular filter paper and placed onto the aforementioned media. The plate was incubated for 2 days at 30˚C and then photographed.

## Luciferase assay

For the real-time luciferase activity assay, cells expressing the pHSE-lucCP+ plasmid [30] were grown to mid-log phase at 25˚C. Activity of Hsf1 was determined by adding luciferin (final concentration 0.5 mm) and distributing 150 μl aliquots of the cultures into a white 96-well plate. Cells were incubated in a Synergy MX Microplate reader (Biotek Instruments) at 37˚C for 200 min, and luminescence was read every 5 min. Graph was prepared using GraphPad Prism 7. All experiments were conducted with at least 5 biological replicates.

## Bacterial expression

BL21 cells transformed with V5-tagged Pim1 and His-Ssa1 were grown to an OD600 of 0.6. V5-tagged Pim1 and His-Ssa1 expression was induced by addition of 1 mM IPTG and incubation of cells at 30˚C for 4 h. Cells were lysed in B-PER reagent (with protease inhibitors) and the lysate was clarified by centrifugation. Protein samples were analyzed using immunoblotting.

## Immunoprecipitation

For FLAG IP, cells were harvested and FLAG-tagged proteins were isolated as follows: Protein was extracted via bead beating in 500 μl binding buffer (50 mM Na-phosphate (pH 8.0), 300 mM NaCl, 0.01% Tween-20). A total of 200 μg of protein extract was incubated with 30 μl anti-Flag M2 magnetic beads (Sigma) at 4˚C overnight. Anti-FLAG M2 beads were collected by magnet then washed 5 times with 500 μl binding buffer. After the final wash, the buffer was aspirated and beads were incubated with 65 μl Elution buffer (binding buffer supplemented with 10 μg/ml 3× FLAG peptide (Apex Bio) for 1 h at room temperature, then beads were collected via magnet. The supernatant containing purified FLAG-protein was transferred to a fresh tube, 25 μl of 4× SDS-PAGE sample buffer was added and the sample was denatured for 5 min at 95˚C. The eluates were separated by SDS-PAGE (7.5% to 10%) and visualized by immunoblotting.

For HA IP, cells were harvested and HA-tagged proteins were isolated as follows: Protein was extracted via bead beating in 500 μl binding buffer (50 mM Na-phosphate (pH 8.0), 300 mM NaCl, 0.01% Tween-20). A total of 200 μg of protein extract was incubated with 30 μl anti-HA magnetic beads (Thermo Fisher Scientific) at 4˚C for 30 min. Anti-HA beads were collected by magnet then washed 5 times with 500 μl binding buffer. After the final wash, the buffer was aspirated and beads were incubated with 65 μl Elution buffer and 15 μl of 4× loading dye and boiled at 100˚C for 10 min, then beads were collected via magnet. The supernatant containing purified HA-protein was transferred to a fresh tube, 25 μl of 4× SDS-PAGE sample buffer was added and the sample was denatured for 5 min at 95˚C. The eluates were separated by SDS-PAGE (7.5% to 10%) and visualized by immunoblotting.

For GFP IP, cells were harvested and GFP-tagged proteins were isolated as follows: Protein was extracted via bead beating in 500 μl binding buffer (50 mM Na-phosphate (pH 8.0), 300 mM NaCl, 0.01% Tween-20). A total of 200 μg of protein extract was incubated with 30 μl

anti-GFP magnetic beads at 4˚C overnight. Anti-GFP beads were collected by magnet then washed 5 times with 500 μl binding buffer. After the final wash, the buffer was aspirated and beads were incubated with 65 μl Elution buffer and 15 ul of 4× loading dye and boiled at 100˚C for 10 min, then beads were collected via magnet. The supernatant containing purified HA-protein was transferred to a fresh tube, 25 μl of 5× SDS-PAGE sample buffer was added, and the sample was denatured for 5 min at 95˚C. The eluates were separated by SDS-PAGE (7.5% to 10%) and visualized by immunoblotting.

## Immunoblotting

For testing the inactivation of Ssa1 function (achieved via the temperature-labile mutant *ssa1-45*), cells were grown to mid-log phase and heat shocked at 39˚C for 4 h. Cell lysates obtained were probed for HA and FLAG antibodies for Hir1, 2, and Mtw1, respectively. Protein extracts were made as described in [90]. Approximately 30 μg of protein was separated by 4% to 12% NuPAGE SDS-PAGE (Thermo Fisher Scientific). Proteins were detected using the primary antibodies incubated with primary antibodies (S2 Table) with the following dilutions: anti-GAPDH (1:5,000), anti-HA-tag (1:2,000), anti-FLAG tag (1:2,000), anti-His tag (1:2,000), anti-GFP tag (1:2,000), anti-Ssa1 (1:2,000), anti-Hsp82 (1:2,000), anti-Sse1 (1:2,000), anti-Hsp90 (1:2,000), anti-Dnaja1 (1:2,000), anti-Hsc70 (1:5,000), anti-Actin (1:5,000), anti-Pgk1 (1:5,000), anti-Tubulin (1:10,000), anti-Lonp-1 (1:2,000). Membranes were washed with TBS-Tween 20 (0.2%) and incubated with the corresponding secondary antibodies: anti-rat IgG-HRP (1:5,000), anti-mouse IgG-HRP (1:5,000), anti-rabbit IgG-HRP (1:5,000), and anti-mouse IgM-HRP (1:5,000). Blots were imaged on a ChemiDoc MP imaging system (Bio-Rad). After treatment with SuperSignal West Pico Chemiluminescent Substrate (Thermo Fisher Scientific). Blots were stripped and re-probed with the relevant antibodies using Restore Western Blot Stripping Buffer (Thermo Fisher Scientific).

HEK293T cells were either un-transfected or transfected with plasmids for expression of Flag, HA, or GFP-tagged proteins using Lipofectamine 3000 (Thermo Fisher Scientific). After 48 h, the cells were washed with 1× PBS and total cell extract was prepared from the cells using M-PER (Thermo Fisher Scientific) containing EDTA-free protease and phosphatase inhibitor cocktail (Thermo Fisher Scientific) according to the manufacturer's recommended protocol. Protein was quantitated using the Coomassie protein assay.

## Protein stability assay

For JG-98 and bortezomib treatments, HEK293T cells were treated with the drugs at indicated concentration and kept in incubator at 37˚C and 5% CO2 for the indicated time points. After each time point, cells were washed with 1× PBS and total cell extracts were prepared using Mammalian Protein Extract Reagent (Thermo Fisher Scientific).

## β-Galactosidase assays

For *Fus1-lacZ* fusion expression experiments, cells were grown overnight in SD-ura media at 30˚C and then re-inoculated at OD600 of 0.2 to 0.4 and then grown for a further 4 h. Cells were treated with 0.5 μm and 5 μm alpha factor for 30 min and then *Fus1-lacZ* fusion assays were carried out as described previously [52]. Briefly, protein was extracted through bead beating and protein was quantitated via Bradford assay. The β-galactosidase reaction containing 100 μg of protein extract in 1 ml Z-Buffer was initiated by addition of 200 μl ONPG (4 mg/ml) and incubated at 28˚C until the appearance of a pale yellow color was noted. The reaction was quenched via the addition of 500 μl $Na_2CO_3$ (1M) solution. The optical density of the reaction was measured at 420 nm. β-Gal activity was calculated using $((OD420 \times 1.7)/(0.0045 \times protein$

× reaction time)), where protein is measured in mg, and time is in minutes. The mean and standard deviation from 3 independent transformants were calculated.

## Fluorescence microscopy

For BiFC experiments, yeast cells were grown to mid-logarithmic phase in SC drop-out media and were examined on a Leica DM6 inverted microscope with an oil immersion objective. Fluorescence images for BiFC were taken using a standard fluorescein isothiocyanate filter set (excitation band pass filter, 450 to 490 nm; beam splitter, 510 nm; emission band pass filter, 515 to 565 nm). For Pim1, cells were cultured in SC complete and YPD for imaging and growth assay, respectively. Gene deletion and fluorescent protein tagging were performed with PCR-mediated homologous recombination and verified by PCR genotyping. Live-cell images were acquired using a Yokogawa CSU-10 spinning disc on the side port of a Carl Zeiss 200 m inverted microscope or a Carl Zeiss LSM-780 confocal system. Imaging quantification was described previously using imageJ [34]. For Mtw1, a Zeiss Axioimager Z2 microscope (Carl Zeiss AG, Germany) was used to image cells using a 63× 1.4NA apochromatic oil immersion lens. Fluorescence was excited using a Zeiss Colibri LED illumination system (GFP = 470 nm, YFP = 505 nm, and RFP = 590 nm) and differential interference contrast (DIC) prisms were used to enhance the contrast in bright field. The emitted light was captured using a Hamamatsu Flash 4.0 Lte CMOS camera with FL-400 (6.5-μm pixels, binned 2 × 2). The exposure time was set to 300 ms to ensure that signal intensities remained below saturation. Images were acquired using the Zen software (Zeiss) and analyzed and prepared using the Icy BioImage Analysis unit (version 2.0.3.0) [91] and FIJI/ImageJ. Fluorescence intensities were quantified using the semi-automated FociQuant ImageJ script [92]. Intensities were compared using the Student $t$ test ($p$-value = $1.8 \times 10^{-12}$).

## Supporting information

**S1 Fig.** (A) Distances between cross-linked amino acids on identified Ssa-Ssa1 peptides based on monomeric Hsp70 in open and close conformations. Maximum allowed cross-link length of DSSO is represented as a dashed line. All cross-linked peptides where the calculated distance between 2 Ssa1 residues is greater than the DSSO spacer arm cannot come from a single Ssa1 molecule. (B) Control for Ssa1-Ssa1 BiFC experiment. Yeast expressing either only VN-Ssa1 or VC-Ssa1 were analyzed for BiFC signal. The data underlying the graphs shown in the figure can be found in S1 Data.
(TIF)

**S2 Fig.** (A) Venn diagram representing previously known physical interactors of Ssa1 versus direct interactors of Ssa1 identified in this study. (B) Scatter plot of number of interactors identified versus surface lysine on the domains of Ssa1. (C) Bar graph representing interactors per cross-linkable lysine on domains of Ssa1. The data underlying the graphs shown in the figure can be found in S1 Data.
(TIF)

**S3 Fig.** (A) Quantification of percentage of cell with mitoFluc labeled DUMP structures in (Fig 5G). (B). Localization of Pim1 wild type and the mutants in yeast cells. Paired $t$ test was used for statistical analysis. (C) Western blot showing the levels of Pim1 wild type and the mutants in yeast cells. The data underlying the graphs shown in the figure can be found in S1 Data.
(TIF)

**S4 Fig.** (A) Growth curves of wild-type (BY4741) and Mtw1-Y86F yeast strains. Cells were diluted from an overnight culture to $OD_{600} = 0.03$ and the $OD_{600}$ was measured every 5 min for 16 h using a microplate reader. Growth analysis was performed in 10 replicates per strain. (B) Doubling times for wild-type and mutant strains were calculated for each replicate and compared using Student $t$ test (mean = 92.97 min, $p$-value = 0.04). The data underlying the graphs shown in the figure can be found in S1 Data.
(TIF)

**S1 Table.** Summary of all mass spectrometry data. Tab 1. All proteins identified in complex with Ssa1 upon DSSO treatment. Tab 2. All cross-linked Ssa1-Ssa1 peptides observed. Tab 3. All cross-linked Ssa1-client peptides observed. Tab 4. All cross-linked client–client peptides observed. Tab 5. All cross-linked peptides detected that contain either a phosphorylation, acetylation, or methylation.
(XLSX)

**S2 Table.** Summary of reagents used. Tab 1. Yeast strains and plasmids used in this study. Tab 2. Summary of major reagents used in this study.
(XLSX)

**S1 Raw Images. Unprocessed images for blots shown in the paper.**
(PPTX)

**S1 Data.** Numerical data for Figs 2F, 5F, 6I, 7D, 7G, S1A, S2B, S2C, S3A, S4A and S4B.
(XLSX)

## Acknowledgments

We thank Dr. Paola Lopez-Duarte for the use of her confocal microscope. Cartoons in Figs 5K, 6K and 7J were created with BioRender.com. We also thank U. Gueldener and J. H. Hegemann for the pUG36 plasmid and J. Bucher for his kind gift of Hsp26 and Hsp42 antibodies.

## Author Contributions

**Conceptualization:** Nitika, Peter H. Thorpe, Andrew W. Truman.

**Data curation:** Nitika, Jake T. Kline, Jacek Sikora, Yuhao Wang, Romain Huguet, Cinzia Klemm, Matthew J. Winters, Rong Li.

**Formal analysis:** Nitika, Linhao Ruan, Siddhi Omkar, Jacek Sikora, Mara Texeira Torres, Jade E. Takakuwa, Kazuo Tatebayashi, Rong Li, Andrew W. Truman.

**Funding acquisition:** Andrew W. Truman.

**Investigation:** Nitika, Bo Zheng, Linhao Ruan, Luca Fornelli, Andrew W. Truman.

**Methodology:** Nitika, Bo Zheng, Linhao Ruan, Jake T. Kline, Siddhi Omkar, Jacek Sikora, Mara Texeira Torres, Romain Huguet, Matthew J. Winters, Peter M. Pryciak, Kazuo Tatebayashi, Andrew W. Truman.

**Resources:** Peter M. Pryciak, Peter H. Thorpe, Andrew W. Truman.

**Software:** Nitika, Linhao Ruan, Jake T. Kline, Jacek Sikora, Mara Texeira Torres.

**Supervision:** Nitika, Andrew W. Truman.

**Validation:** Nitika, Bo Zheng.

**Writing – original draft:** Nitika, Bo Zheng, Verónica A. Segarra, Peter M. Pryciak, Peter H. Thorpe, Luca Fornelli, Andrew W. Truman.

**Writing – review & editing:** Nitika, Siddhi Omkar, Andrew W. Truman.

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
