## [Editor Report · Decision Letter 0]

11 Mar 2022

Dear Dr Truman, 

Thank you for submitting your manuscript entitled "A novel multifunctional role for Hsp70 in binding post-translational modifications on client proteins" for consideration as a Research Article by PLOS Biology.

Your manuscript has now been evaluated by the PLOS Biology editorial staff, as well as by an academic editor with relevant expertise, and I am writing to let you know that we would like to send your submission out for external peer review. We would like to consider your submission as a 'Methods and Resources' article (more Resource than Method), so we ask that you please tick this as the article type upon resubmission (see below).

Before we can send your manuscript to reviewers, we need you to complete your submission by providing the metadata that is required for full assessment. To this end, please login to Editorial Manager where you will find the paper in the 'Submissions Needing Revisions' folder on your homepage. Please click 'Revise Submission' from the Action Links and complete all additional questions in the submission questionnaire.

Once your full submission is complete, your paper will undergo a series of checks in preparation for peer review. Once your manuscript has passed the checks it will be sent out for review. To provide the metadata for your submission, please Login to Editorial Manager (https://www.editorialmanager.com/pbiology) within two working days, i.e. by Mar 13 2022 11:59PM.

If your manuscript has been previously reviewed at another journal, PLOS Biology is willing to work with those reviews in order to avoid re-starting the process. Submission of the previous reviews is entirely optional and our ability to use them effectively will depend on the willingness of the previous journal to confirm the content of the reports and share the reviewer identities. Please note that we reserve the right to invite additional reviewers if we consider that additional/independent reviewers are needed, although we aim to avoid this as far as possible. In our experience, working with previous reviews does save time. 

If you would like to send previous reviewer reports to us, please email me at rhodge@plos.org to let me know, including the name of the previous journal and the manuscript ID the study was given, as well as attaching a point-by-point response to reviewers that details how you have or plan to address the reviewers' concerns. 

Given the disruptions resulting from the ongoing COVID-19 pandemic, please expect some delays in the editorial process. We apologise in advance for any inconvenience caused and will do our best to minimize impact as far as possible.

Kind regards,

Richard

Richard Hodge, PhD

Associate Editor, PLOS Biology

rhodge@plos.org

PLOS

---

## [Decision Letter · Decision Letter 1]

15 Apr 2022

Dear Dr Truman,

Thank you for submitting your manuscript "A novel multifunctional role for Hsp70 in binding post-translational modifications on client proteins" for consideration as a Methods and Resources article at PLOS Biology. Please accept my apologies for the delays you have experienced during the peer review process. Your manuscript has been evaluated by the PLOS Biology editors, an Academic Editor with relevant expertise, and by three independent reviewers.

The reviews are attached below. You will see that the reviewers find your manuscript interesting and useful for the field, but raise overlapping concerns with the XL-MS approach and whether the interactions are physiologically relevant or an artefact due to the use of the cross-linker. In addition, they highlight the lack of reporting details and quantifications for the pulldown data. After discussions with the academic editor, we agree that additional experiments should be included to validate the specificity of the method, such as using purified proteins with NBD fragments or further analysing Ssa1 dimerization with mutant constructs. 

In light of the reviews, we will not be able to accept the current version of the manuscript, but we would welcome re-submission of a much-revised version that takes into account the reviewers' comments. We cannot make any decision about publication until we have seen the revised manuscript and your response to the reviewers' comments. Your revised manuscript is also likely to be sent for further evaluation by the reviewers.

We expect to receive your revised manuscript within 3 months. Please email us (plosbiology@plos.org) if you have any questions or concerns, or would like to request an extension. At this stage, your manuscript remains formally under active consideration at our journal; please notify us by email if you do not intend to submit a revision so that we may end consideration of the manuscript at PLOS Biology.

**IMPORTANT - SUBMITTING YOUR REVISION**

*Re-submission Checklist*

*Published Peer Review*

*PLOS Data Policy*

*Blot and Gel Data Policy*

Sincerely,

Richard

Richard Hodge, PhD

Associate Editor, PLOS Biology

rhodge@plos.org

REVIEWS:

Reviewer #1: Hsp70 is the predominant molecular chaperone, known to play many roles in protein synthesis, assembly, regulation, and turnover. Defining the client and partner interactomes of this key cellular component has been challenging due to the nature of the majority of these interactions being low-affinity and transient. In this manuscript, the authors utilize lysine protein crosslinking in cocnert with mass spectrometry to catalog Hsp70 interactors. Notably, the approach provides compelling evidence, backed up by additional experiments, that the yeast Hsp70 Ssa1 dimerizes in vivo. Because the peptides also provide detailed molecular signatures, it was also possible to identify and test the relevance of post-translational modifications of putative clients, and several different cases are validated and explored in greater depth: the histone regulatory complex Hir1/2, the mitochondrial matrix protease Pim1, the kinetochore protein Mtw1, and the signaling kinase Ste11. In all cases, PTMs appear to influence association with Hsp70, and the chaperone is required for stability and function of the client. Together, this study demonstrates the utility of XL-MS for identifying transient interactomes, provides novel insight into Hsp70 behavior and interactions, and suppports models wherein client PTM are dependent on or dictate chaperone involvement.

This is an intriguing report that breaks new ground in study of the Hsp70 chaperone, exploiting XL-MS to make several new discoveries. Importantly, evidence is provided that not only bolsters the identification of these new clients in yeast but also validates them in mammalian cell lines. While the assessment of the individual clients is limited in scope, the experiments significantly and mostly satisfactorily support the proteomic approach. A major finding that remains puzzling is the large number of interactors that bind to the NBD or SBDalpha, including all of the clients that received detailed attention. It is difficult to square these interactions as traditional Hsp70 clients, but not without precedent (see reference (and consider including) Tripathi, A., Mandon, E. C., Gilmore, R., and Rapoport, T. A. (2017) Two alternative binding mechanisms connect the protein translocation Sec71-Sec72 complex with heat shock proteins. J. Biol. Chem. 292, 8007-8018). Like the Ssa1-Ssa1 interaction, these findings are dogma-challenging but not fully explored or explained due to the broad nature of the report. While my enthusiasm for the study is high, parts of the analysis lack rigor and are not as compelling as interpreted by the authors. Additional work and/or clarification is therefore needed to fully support the conclusions made in the paper.

1. Fig.1 - the purported nuclear dimer of Ssa1 is very intriguing and supported by the co-IP and BiFC experiments. However, the postulated model in panel 1C seems inconsistent with the multitutude of crosslinks represented in 1D. This raises several possibilities - i) the XL is potentially artifactual, ii) the XL represents not dimers but higher-order assemblies that would allow packing of all domains within spacer arm distance of each other, iii) the crystal structure in 1C is inappropriate as a model for Ssa1-Ssa1 dimerization. Extraordinary claims require extraordinary proof, and this isn't quite it. A major absence is the experiment showing that the E540A/N537K mutants do not dimerize - this is inferred but not tested despite the reagents being in hand. 

2. Fig. 1G - I believe the N537A nomenclature is incorrect as the results text reads N537K. 

3. Fig. 1G and 1H - the temperature sensitive phenotype is indeed quite dramatic and an attempt is made to link the growth phenotype to a reduced HSR. While this is possible, it is not consistent with past results that suggest that impaired HSR does not typically result in ts growth but rather heat shock sensitivity, which could be easily tested. More importantly, it would be useful to assess Hsf1 target protein levels in the strains to ascertain if chaperone levels are truly significantly reduced in the mutant strains. I suspect they are not. Finally, the corresponding author is well aware that chaperone and Hsf1 mutants are typically ts due to failure to maintain the cell wall stress response. This should also have been tested using sorbitol plates as done later in the manuscript to ask whether osmotic support suppresses the ts phenotype.

4. Fig. 1I - why is the load control protein GAPDH present in the FLAG-Ssa1 IP? This outcome casts doubt on the quality of the IP and the veracity of the pulldowns in general.

5. As an extension of comment #4, the paper relies heavily on differential signals by immunoblot for protein levels and pulldowns, yet no quantitation of these blots is provided. Many of the lysate control lanes exhibit significant variability in protein signal and the differences are not immediately obvious in several cases. It is imperative that replicate experiments be quantitated with error ranges shown to raise confidence in the data.

6. Fig. 3 C - what is the purpose of the cartoon of a yeast cell aboce the pulldowns? What is happening in the lysates for Hsp82? 

7. Fig. 4F - the JG-98 + bortezomib blots are highly overexposed, making it difficult to discern whether the proteasome inhibitor has truly blocked HIRA-HA degradation. Lighter exposures and band quantitation would remedy this.

8. Fig. 5D - what is being pulled down here - the results say Lonp-1 is the bait while the figure suggests it is FLAG-Hsc70.

9. It is unclear what is actually happening in Fig. 5H and 5J. Does bortezomib inhibit the lon protease? If so, a citation would be useful to support that fact. Why is the maturation efficiency of protein processing so poor? SDS-PAGE siae markers would also be useful for this panel. In 5J, the results state that self-cleavage was restored in the S974D mutant. However, it appears that cleavage is actually blocked in the asparate mutant in the presence of Ssa1 - am I misinterpreting the banding patterns here?

10. With regard to the entire interaction story for Pim1, this protein is a mitochodrial matrix protein that presumably is not processed until after delivery through the translocons. Is Ssa1 associated with cytosolic Pim1 prior to mitocondrial insertion? No comments are made regarding the organellar nature of this client. 

11. Fig. 7 - much of the results in this figure appear to show no real effects of the mutations and should either be moved to the SI or the authors should consider removing the figure entirely - there's not a lot going on here and it ends the paper on a less-than-compelling note.

12. Why is the presence of Hsp82 investigated in most if not all of the interactions? The positive results here suggest that these are actually clients of the greater Hsp90 system, which of course include Hsp70 as a precursor step. It is also unusual for both Hsp90 and Hsp70/Ydj1 to be present at the same time, so the pulldowns likely represent mixed populations of clients with several different chaperones.

Reviewer #2: Nitika et al provide an in vivo interactive analysis of the Hsp70 chaperone Ssa1, using cross-linking mass spectrometry. If correct, this study is absolutely awesome. The identify hundreds of new interactions, they observe Hsp70 dimerisation in vivo and phosphorylation of Hsp70 interactions. My main concern is that I wonder whether the basis of the study, the XL-MS analysis really reveals reliable data.

Some examples of points that questions the credibility of the approach:

1. Almost 80% of the interactions are located in the nucleotide binding domain (NBD). While it is plausible that regulatory factors interact with the NBD, it not plausible that such interactions outnumber substrate interactions by 4:1. In fact, I would have even expected more interactions in the substrate binding domain than identified by the authors. This raises the questions whether the interactions monitored here are relevant and really reflect meaningful interactions inside the cell. Ssa1 is a rather abundant protein, equipped with a cross linker it may interact with many proteins. Would overproduction of NBD lead to a similar interactive? Would it me toxic, by competing out the interactions? Would V435F be toxic? How does the interactive of this mutant compare to wildtype?

2. The authors find many interactions for the dimerisation of Hsp70. When analysing substrate interactions, they only observe single cross-links. This suggests that concentration indeed plays a major role for the findings of the study. E.g. Hir1 shows only 1 cross link to the SBD, Pim1 only 1 cross link to the NBD.

3. Pim1 is a mitochondrial protease. Why should it now interact with the NBD of Hsp70?

4. For the analysis of dimerisation, the question is to which extent does dimerisation play a role - how much of Hsp70 is present in the dimer? if the interaction is meaningful under stress, what would be its physiological role? Dimerisation had been suspected in the literature as a storage function (although never proven in vivo), but this would expect that under heat stress Hsp70 should more monomers to interact with unfolded clients.

Credit to the authors that they provide several findings of the study with additional data. To a large extent this is based on Ip data, which are not quantitative and prone to indirect interactions. If the authors are right they would revise many paradigms. However, at present I am not convinced. I am hesitant to ask the authors to provide more data on the individual interactions, it would be an incredible amount of work and would needed to be published in several papers. I could not come up with the killer experiment that would convince me. I hope the questions above would inspire the authors come up with data that would satisfy my concerns.

Reviewer #3: The manuscript describes the identification of the Hsp70 interactome from yeast cells using a combination of crosslinking and mass spectrometry. The authors expressed a tagged version of yeast Hsp70 (Ssa1) in Ssa1-4 null yeast and subsequently used affinity purification to isolate protein complexes in the presence or absence of an MS compatible crosslinker DSSO. In addition, the authors can identify novel posttranslational modifications on the proteins in the complex. The authors subsequently use orthogonal methods to analyze aspects of Ssa1 function, including dimerization, and to validate the interaction of selected novel interactors and in some cases the PTML with Ssa1 and to provide some preliminary insight as to the client status of these proteins and the putative roles of Ssa1 in these complexes. 

As a submission under the 'methods and resources' section, the manuscript describes both a method (the XL-MS analysis) and provides a resource dataset (the interactome, consisting of total interactome, direct binders and posttranslational modifications). In particular, the ability to simultaneously analyze indirect and direct interactors on a global scale is powerful. In particular, the sharing of both the raw data files and selected synthesized data (in the form of excel spreadsheets) makes the data widely accessible. The dataset is applicable to studying Hsp70 function but has broader application since novel PTLMs were identified and can be useful independently of Hsp70, and the crosslinked peptides between non-Hsp70 peptides is useful in wider analysis of direct and indirect protein complexes. The method as described is applicable to other protein-protein interactions and the datasets provide global data containing novel observations that can form the basis of future mechanistic study by both this group and others.

In addition to the method and resources, the authors conduct some additional analyses to demonstrate biological insight which also serves to validate the approach. They selected 4 novel interactors and demonstrate Hsp70 interaction, Hsp70 client status, conservation in human cells and preliminary functional role. The results raise some interesting conceptual questions about Hsp70 function in the cell. Indeed, the data presented may suggest that the Hsp70 interactome is more specific than previously appreciated and that the functions of Hsp70 extend beyond just protein folding. The manuscript was written in a readable and accessible style, and the data are beautifully and clearly presented. 

I have some minor suggestions for the authors to consider during revision of the manuscript:

Figure 1A describes the core of the methodology but is hardly mentioned in the narrative at all. It is left implicit that the reader will figure out the approach from the figure. It would be useful to the reader to provide a concise but informative narrative to describe the broad approach in the text. 

Fig 2F: there are no controls for the BiFc analysis. As a minimum, the untagged VN and VC should be shown for the reader to have confidence in the data. 

Fig 2G-H: I am not clear on why the authors selected these experiments to demonstrate dimerisation. I understand that dimerisation is required for function DnaK activity, but the growth phenotype and HSE response with the mutant does not specifically demonstrate lack of dimer formation. The growth phenotype could arise due to another functional role for that residue. Would it be possible to conclusively demonstrate loss of dimerisation using the IP or BiFc approach as reported by the authors for example? 

The authors use the term 'client' quite loosely when describing the interactome analysis. My understanding, which is reinforced by the authors themselves later in the ms, is that interaction alone is not sufficient to infer client status. A client is one that interacts and is reliant on the chaperone (i.e. is perturbed upon chaperone inhibition). Therefore, it would be more accurate to use the term "interactor" or similar unless where the authors demonstrate client status. 

Fig 4 D: the - and + signs are not correctly aligned with the lanes

Figure 5H and I: should it be Pim-GFP?

Fig 5I: there is reference to a significant increase in the narrative but no stats are shown

Fig 6H and I: How certain are you at the increased localization to the kinetochore of the YFP tagged protein is not just due to higher levels of protein expression? There did appear to be high levels of diffuse fluorescence in the background?

Page 14, line 5: 'previous' should be 'previously'

The methods section needs to be reviewed for formatting and consistency. There are several minor issues that should be easily resolved. These include: 

CO2 should have the 2 as subscript

E. coli should be in italics

Media should be medium unless when referring to different types of media

His tagged or His-tagged - please be consistent

150ug should be 150 µg

Spacing issues in the methods e.g. "1MNH4HCO3" and also cases where there are sometimes spaces between units and numbers and instances where there are not

"re-suspended-in" should be "re-suspended in" 

Check the wording of this sentence please as it does not appear complete "Gene ontology analysis of DSSO treated Ssa1 immunoprecipitated complexes and crosslinked Ssa1 complexes using TheCellMap.org." 

In legends "Scale bars are 10 μM". Should be 10 µm

Cross links versus cross-links - please be consistent

Legend for Figure 7: the descriptions for panels A and B are switched - please check

---

## [Decision Letter · Decision Letter 2]

14 Sep 2022

Dear Andy,

Thank you for your patience while we considered your revised manuscript "Cross-linking mass spectrometry analysis of Hsp70 complexes reveal novel PTM-associated interactions" for publication as a Methods and Resources Article at PLOS Biology. I am very sorry for the delay in getting back to you with our decision and I hope this letter arrives in time before your upcoming grant deadlines. This revised version of your manuscript has been evaluated by the PLOS Biology editors, the Academic Editor and the original reviewers.

Based on the reviews, I am pleased to say we are likely to accept this manuscript for publication, provided you satisfactorily address the remaining data and other policy-related requests that I have provided below (A-I):

(A) We would like to suggest the following modification to the title:

“Comprehensive characterization of the Hsp70 interactome reveals novel client proteins and interactions mediated by post-translational modifications”

(B) Please define ‘PTMs’ when used in the Abstract in the first instance.

(C) You may be aware of the PLOS Data Policy, which requires that all data be made available without restriction: http://journals.plos.org/plosbiology/s/data-availability. For more information, please also see this editorial: http://dx.doi.org/10.1371/journal.pbio.1001797

Figure 2F, 5F, 6I, 7D, 7G, S1A, S2B-C, S3A, S4A-B 

(D) Please also ensure that each of the relevant figure legends in your manuscript include information on *WHERE THE UNDERLYING DATA CAN BE FOUND*, and ensure your supplemental data file/s has a legend.

(E) Thank you for depositing the raw mass spectrometry data in the MassIVE database. However, I note that the link to access the files can only be opened with an app and I am unable to access it. The data is also password protected. Would it be possible to deposit this data with the ProteomeXchange consortium directly and then provide the accession number in the Data Statement in the online submission form? This will make it easier for readers to access the data. 

(F) Please ensure that your Data Statement in the submission system accurately describes where your data can be found and is in final format, as it will be published as written there.

(G) Thank you for already providing the raw and original blot/gel images in the Supplementary Information. However, I note that the images for the following figures are missing in the file:

Figure 2G, 5J, 7E, S3C

In addition, I note that not all of the images provided are fully uncropped. Please carefully read our guidelines for how to prepare and upload this data: https://journals.plos.org/plosbiology/s/figures#loc-blot-and-gel-reporting-requirements .

I would be grateful if you could provide the fully uncropped images if they are available - if they were cut during the blotting process then this is OK.

(H) Please note that per journal policy, the model system/species studied should be clearly stated in the abstract of your manuscript. 

(I) Please also provide a blurb which (if accepted) will be included in our weekly and monthly Electronic Table of Contents, sent out to readers of PLOS Biology, and may be used to promote your article in social media. The blurb should be about 30-40 words long and is subject to editorial changes. It should, without exaggeration, entice people to read your manuscript. It should not be redundant with the title and should not contain acronyms or abbreviations. For examples, view our author guidelines: https://journals.plos.org/plosbiology/s/revising-your-manuscript#loc-blurb

*Published Peer Review History*

*Press*

Best wishes,

Richard

Richard Hodge, PhD

Associate Editor, PLOS Biology

rhodge@plos.org

Reviewer remarks:

Reviewer #1 (Kevin Morano, signs review): The authors have done a terrific job addressing the vast majority of the concerns raised by all the reviewers. As they point out, the work is intended to be a Resource article and as such, the results and conclusions are consistent with that class of publication in PLoS Biology. This paper will spur much discussion in the field. No further issues.

Reviewer #2: The authors sufficiently addressed all comments made by this reviewer.

Reviewer #3: Thank you to the authors for their careful review of the manuscript. I am satisfied that all of my comments have been address appropriately and I am happy to recommend acceptance of this manuscript. It provides an approach that will be useful for mapping global protein-protein interactions and the associated dataset will support important fundamental studies on the regulation of Hsp70-mediated proteostasis in future.

---

## [Editor Report · Decision Letter 3]

21 Sep 2022

Dear Andy,

On behalf of my colleagues and the Academic Editor, Ursula Jakob, I am pleased to say that we can accept your manuscript for publication, provided you address any remaining formatting and reporting issues. These will be detailed in an email you should receive within 2-3 business days from our colleagues in the journal operations team; no action is required from you until then. Please note that we will not be able to formally accept your manuscript and schedule it for publication until you have completed any requested changes.

PRESS

Best wishes, 

Richard

Richard Hodge, PhD

Associate Editor, PLOS Biology

rhodge@plos.org

PLOS
